# HoPE: Hybrid of Position Embedding for Long Context Vision-Language Models

**Haoran Li**[1]   **Yingjie Qin**[2]   **Baoyuan Ou**[2]   **Lai Xu**[2]   **Ruiwen Xu**[2]

[1]Carnegie Mellon University    [2]Xiaohongshu Inc.

`haoranl4@cs.cmu.edu`

## Abstract

Vision-Language Models (VLMs) have made significant progress in multimodal tasks. However, their performance often deteriorates in long-context scenarios, particularly long videos. While Rotary Position Embedding (RoPE) has been widely adopted for length generalization in Large Language Models (LLMs), extending vanilla RoPE to capture the intricate spatial-temporal dependencies in videos remains an unsolved challenge. Existing methods typically allocate different frequencies within RoPE to encode 3D positional information. However, these allocation strategies mainly rely on heuristics, lacking in-depth theoretical analysis. In this paper, we first study how different allocation strategies impact the long-context capabilities of VLMs. Our analysis reveals that current multimodal RoPEs fail to reliably capture semantic similarities over extended contexts. To address this issue, we propose HoPE, a Hybrid of Position Embedding designed to improve the long-context capabilities of VLMs. HoPE introduces a hybrid frequency allocation strategy for reliable semantic modeling over arbitrarily long contexts, and a dynamic temporal scaling mechanism to facilitate robust learning and flexible inference across diverse context lengths. Extensive experiments across four video benchmarks on long video understanding and retrieval tasks demonstrate that HoPE consistently outperforms existing methods, confirming its effectiveness. Our code is available at `https://github.com/hrlics/HoPE`.

## 1 Introduction

Vision-Language Models (VLMs) [1–5] have achieved remarkable success in multimodal tasks, including visual question answering [6–9], image captioning [10, 11], cross-modal retrieval [12, 13], and more [14–16]. However, VLMs suffer from significant performance degradation in long-context scenarios, particularly long videos [17–20]. In such settings, VLMs even struggle with simple tasks like object counting and temporal localization [21, 22], revealing a critical limitation in their ability to model extended spatial-temporal dependencies. This limitation substantially hinders their real-world deployment, where input length often exceeds the context window they have been pretrained on.

Rotary Position Embedding (RoPE) [23] has been widely adopted for length generalization in text-based LLMs [24–26]. Specifically, RoPE incorporates positional information by partitioning the query and key vectors into 2-dimensional pairs and rotating each pair at a unique frequency that decreases as the dimensional index increases. Despite its advantages, directly applying 1D RoPE fails to capture the intricate spatial-temporal dependencies in videos. Several methods have been proposed to extend 1D RoPE for multimodal inputs [2, 27, 28]. Among these, the most common approach is to allocate different frequencies to encode different positional components, as shown in the upper plots of Figure 1. For example, M-RoPE [2] allocates the *highest* frequencies for temporal modeling ($t$), and the remaining low frequencies for spatial modeling ($x, y$). In contrast, VideoRoPE [28] proposes to allocate the *lowest* frequencies for temporal dimensions ($t$), and further

39th Conference on Neural Information Processing Systems (NeurIPS 2025).

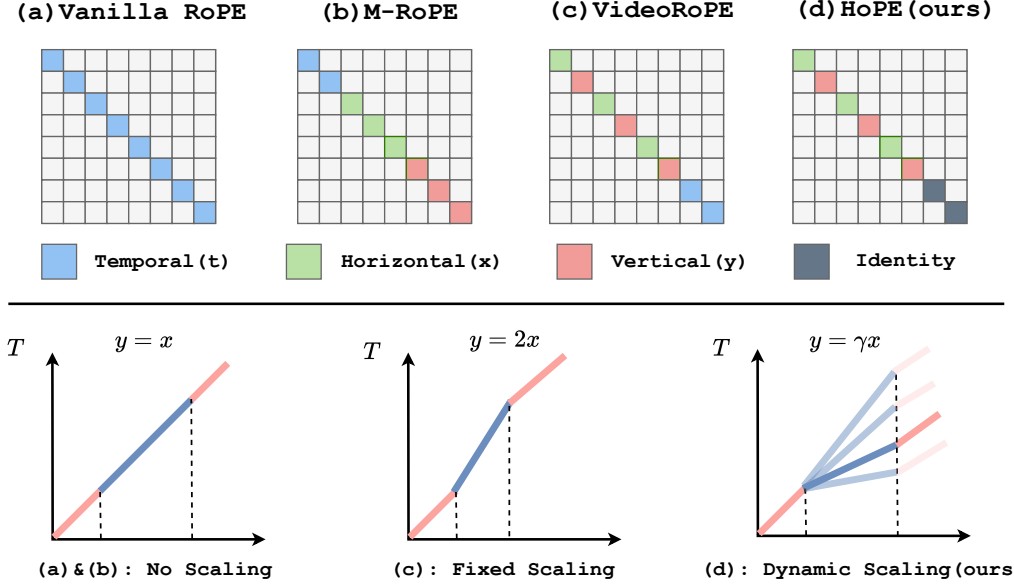

Figure 1: **Comparison of our HoPE and existing methods.** Upper plots illustrate the frequency allocation strategies in different RoPE variants. Here, frequency decreases along the diagonal. (d) HoPE sets the lowest frequencies to zero for reliable long-range semantic modeling. Lower plots demonstrate different temporal scaling mechanisms. (d) HoPE proposes dynamic and bidirectional scaling to learn temporal dynamics at multiple scales, facilitating robustness to various video speeds.

applies a fixed scaling factor to scale the temporal indices of visual tokens, as shown in the lower plots of Figure 1. Despite their improved performance, two significant challenges remain unsolved. Firstly, current methods mainly rely on heuristics rather than theoretical analysis to determine the frequency allocation strategy. Second, applying a fixed and unidirectional scaling factor for all videos is suboptimal in real-world scenarios, where videos proceed at different speeds and demonstrate significant variance in information densities.

To address these challenges, we begin with an in-depth theoretical analysis in Section 3, outlining the ideal properties that a multimodal RoPE should possess. Our analysis reveals that: (1) the flattening operation in vanilla RoPE inherently violates spatial-temporal locality prior, which is crucial in video modeling; (2) despite diverse frequency allocation strategies, existing multimodal RoPE variants fail to reliably capture semantic similarities over extended contexts; (3) temporal scaling of visual tokens should include both compression and expansion to accommodate varying video speeds in real-world scenarios. Guided by these insights, we propose HoPE, a **H**ybrid **o**f **P**osition **E**mbedding designed to improve the long-context capabilities of VLMs. As shown in Figure 1, HoPE first introduces a hybrid frequency allocation strategy to facilitate long-range semantic modeling. In this strategy, higher frequencies, which are more sensitive to positional differences and better at capturing local features, are allocated to spatial components $(x, y)$ in an interleaved manner. Meanwhile, the lower frequencies are directly set to zero and allocated to temporal component $(t)$ to enable reliable semantic modeling. Moreover, HoPE develops a dynamic temporal scaling mechanism for lengthy visual tokens. This mechanism not only enhances VLMs' robustness to various video speeds, which are common in real-world scenarios, but also offers flexible scaling during inference across diverse context lengths.

We summarize our contributions as follows:

- To our best knowledge, we provide the first theoretical analysis of how different frequency allocation strategies in multimodal RoPEs impact the long-context capabilities of VLMs, offering insights for the design and analysis of future multimodal RoPEs.
- Guided by our analysis, we propose HoPE, which consists of a hybrid frequency allocation strategy for reliable semantic modeling in long-context scenarios, and a dynamic temporal scaling mechanism for robust and flexible temporal comprehension.
- Extensive experiments on four video benchmarks demonstrate that HoPE consistently outperforms existing RoPE variants, achieving improvements of 22.23% and 8.35% on long video retrieval and long video understanding tasks, confirming its effectiveness.

## 2 Preliminaries

**Rotary Position Embedding (RoPE).** Current Transformer-based LLMs rely on Positional Encodings (PEs) to incorporate sequential information into the attention mechanism. Among various PEs, Rotary Position Embedding (RoPE) [23] has emerged as a dominant approach for long-context modeling in text-based LLMs [24–26]. The key to RoPE's success lies in its ability to encode *relative* position information through an *absolute* positional encoding approach, ensuring both effectiveness and efficiency. Consider query and key vectors with $d$ dimensions (where $d$ is an even number), RoPE partitions the dimensions into $d/2$ pairs, e.g., $\mathbf{q}_n = [\mathbf{q}_n^{(0)}; \mathbf{q}_n^{(1)}; \ldots; \mathbf{q}_n^{(d/2-1)}]$. Each pair of dimensions is assigned a unique rotation angle $\theta_i = b^{-2i/d}, i \in \{0, 1, \ldots, d/2 - 1\}$, where $b$ is a pre-defined frequency base and set to $10{,}000$ by default [23]. This rotation can be achieved through a rotation matrix as follows:

$$r^{(i)} = \begin{pmatrix} \cos\theta_i & -\sin\theta_i \\ \sin\theta_i & \cos\theta_i \end{pmatrix}. \tag{1}$$

The overall rotation matrix $\mathbf{R}_n$ is constructed by concatenating each pair's rotation matrix along the diagonal to form a block-diagonal matrix, i.e., $\mathbf{R}_n = \mathrm{diag}(r^{(0)}, r^{(1)}, \ldots, r^{(d/2-1)}) \in \mathbb{R}^{d \times d}$. Therefore, during attention computation, the attention score[1] $\mathbf{A}_{n,m}$ between the $n$-th query $\mathbf{q}_n$ and $m$-th key $\mathbf{k}_m$ is:

$$\mathbf{A}_{n,m} = (\mathbf{q}_n\mathbf{R}_n)(\mathbf{k}_m\mathbf{R}_m)^\top = \mathbf{q}_n\mathbf{R}_{n-m}\mathbf{k}_m^\top, \tag{2}$$

where the rotation matrix $\mathbf{R}_{n-m}$ can be formulated as:

$$\mathbf{R}_{n-m} = \begin{pmatrix} \cos\theta_0(n-m) & -\sin\theta_0(n-m) & \cdots & 0 & 0 \\ \sin\theta_0(n-m) & \cos\theta_0(n-m) & \cdots & 0 & 0 \\ \vdots & \vdots & \ddots & \vdots & \vdots \\ 0 & 0 & \cdots & \cos\theta_{(d/2-1)}(n-m) & -\sin\theta_{(d/2-1)}(n-m) \\ 0 & 0 & \cdots & \sin\theta_{(d/2-1)}(n-m) & \cos\theta_{(d/2-1)}(n-m) \end{pmatrix}. \tag{3}$$

It can be observed that through pairwise attention computation, the final rotation matrix naturally incorporates the *relative* position information $(n - m)$ between the query-key pair.

**No Positional Encoding (NoPE).** Despite the popularity of RoPE, several works have pointed out that the *causal* attention mechanism in current decoder-only LLMs implicitly learns *absolute* positional information [30–32]. This motivates the development of No Positional Encoding (NoPE). Specifically, the *causal* attention mask enforces $A_{n,m} = 0$ for all $m > n$, ensuring that each token only attends to itself and previous tokens. Under this constraint, the attention score with NoPE is a simple dot product between the query vector $\mathbf{q}_n$ and the key vector $\mathbf{k}_m$, i.e., $\mathbf{A}_{n,m} = \mathbf{q}_n\mathbf{k}_m^\top$, providing no explicit positional information to Transformers.

## 3 Analysis

In this section, we conduct a comprehensive theoretical analysis of multimodal RoPE variants, aiming to answer the following questions: **(1)** Is vanilla RoPE enough for long-context VLMs? **(2)** How do different frequency allocation strategies impact semantic modeling in long-range multimodal contexts? **(3)** How should we assign the temporal indices for text and visual tokens?

### 3.1 Vanilla RoPE Fails in Spatial-Temporal Structure

Several recent VLMs [1, 5, 20, 33–35] still use vanilla RoPE for multimodal inputs. In their approach, each video frame is first encoded by a vision encoder (e.g., ViT [36]) and then flattened into a sequence of patch-level tokens. These visual tokens will be treated equally as text tokens for

---

[1]Here, we omit the softmax function and $1/\sqrt{d}$ scaling in standard Transformer [29] for simplicity.

positional encoding, with each token incorporating only 1D temporal information. We show in Proposition 3.1 that this approach, while easy to implement, distorts spatial-temporal localities and fundamentally limits VLMs' ability to model extended spatial-temporal dependencies.

**Proposition 3.1** (1D RoPE violates spatial-temporal locality priors). *Given any query* $\mathbf{q}$ *at position* $(t, x, y)$ *and a relative distance of 1 in spatial or temporal dimensions, the flattening operation in 1D RoPE distorts the relative distance with a magnitude dependent on the frame resolution.*

We provide the proof in Appendix A.1. This mismatch between positional encoding and the 3D structure of videos creates distorted attention patterns, making it difficult for models to learn meaningful spatial-temporal relationships essential for video-related tasks.

> **Conclusion 1.** Directly applying vanilla RoPE to multimodal long-context inputs inherently fails to capture their complex spatial-temporal dependencies.

## 3.2 Current Multimodal RoPEs Are Unreliable in Long-Range Semantic Modeling

To capture the spatial-temporal structure of multimodal inputs, a recent VLM, Qwen2-VL [2], has introduced a Multimodal Rotary Position Embedding (M-RoPE). Concretely, M-RoPE partitions the 128-dimensional rotary embedding into three distinct groups: the first 32 dimensions for temporal information ($t$), the subsequent 48 dimensions for horizontal spatial information ($x$), and the final 48 dimensions for vertical spatial information ($y$), i.e., $\mathbf{R}_{t,x,y} = \mathrm{diag}(\mathbf{R}_t, \mathbf{R}_x, \mathbf{R}_y)$. While this approach realizes a naive extension for RoPE, a fundamental question remains to be answered:

*How do different frequency allocation strategies impact the performance of multimodal RoPE?*

This question arises from the fact that in RoPE, different dimensions carry unique frequencies ($\theta_i = b^{-2i/d}$, $i \in \{0, 1, \ldots, d/2 - 1\}$), as shown in Equation 3. Therefore, different strategies exist for frequency allocation in multimodal RoPE. As shown in Figure 1, M-RoPE allocates the highest frequencies for $t$, intermediate frequencies for $x$, and the lowest frequencies for $y$. In contrast, VideoRoPE [28] proposes to assign the lowest frequencies to temporal modeling ($t$) and high frequencies to spatial dimensions ($x, y$). Their empirical justification stems from attention pattern analysis, which reveals that dimensions encoded with the lowest frequencies exhibit a more pronounced *attention sink* phenomenon [37], which has proven to be effective in long-context modeling. However, we argue that using the lowest frequencies for temporal modeling is still unreliable in capturing semantic similarities in extended multimodal contexts. Specifically, we first introduce semantic preference, a property where attention mechanisms should prioritize semantically similar tokens regardless of their relative distance, and formally define this concept in Definition 3.1.

**Definition 3.1** (Semantic Preference). For any query vector $\mathbf{q}$ and a semantically similar key vector $\mathbf{k}'$ that can be expressed as $\mathbf{k}' = \mathbf{q} + \delta$ where $\delta$ is a zero-mean perturbation, the attention score with RoPE should satisfy:

$$\mathbb{E}_{\mathbf{q},\mathbf{k},\delta}[\mathbf{q}\mathbf{R}_{\Delta t \Delta x \Delta y}\mathbf{k}'^{\top} - \mathbf{q}\mathbf{R}_{\Delta t \Delta x \Delta y}\mathbf{k}^{\top}] \geq 0,$$

where $\mathbf{k}$ is the key vector of a semantically unrelated token. This preference should hold regardless of the relative distance ($\Delta t, \Delta x, \Delta y$) between the query and key.

Then, we show in Theorem 3.1 that ***all*** frequency allocation strategies of current multimodal RoPEs, including selecting the highest/lowest frequencies for temporal modeling, are unreliable in maintaining the semantic preference property over extended contexts. This limitation arises because, with sufficiently long contexts, even the lowest frequencies can produce arbitrary rotations, ultimately undermining semantic preference. We provide the proof in Appendix A.2.

**Theorem 3.1.** *Let* $X = [x_1, x_2, \ldots, x_L]$ *be an input sequence, and let RoPE use any fixed set of temporal frequencies (e.g., highest or lowest). Then there exists a critical length* $L_c$ *such that for all* $L \geq L_c$, *the semantic preference property (Definition 3.1) is violated.*

> **Conclusion 2.** There exist different frequency allocation strategies to extend vanilla RoPE to multimodal RoPE. However, we prove that none of these strategies can reliably maintain the semantic preference property over a sufficiently long context.

### 3.3 How to Assign Positional Index for Multimodal Inputs?

Currently, most VLMs [1–4, 19, 20] adopt the same temporal stride for video frames and text tokens, as shown in Figure 1. However, this approach overlooks the inherent difference in information densities between text and visual tokens. To address this issue, VideoRoPE [28] applies a fixed scaling factor (implemented as 2) to adjust the temporal indices of visual tokens, achieving better empirical performance. However, this rigid scaling approach lacks the flexibility needed for diverse real-world videos, which naturally vary in pace and information density. A more ideal approach would incorporate both temporal compression and expansion capabilities, allowing the model to learn multi-scale temporal relationships, thereby enabling more robust temporal modeling.

> **Conclusion 3.** Temporal index scaling of visual tokens is crucial for balancing multimodal information, yet current methods lack flexibility and bidirectionality.

## 4 HoPE: Hybrid of Position Embedding for Long Context VLMs

To address the above challenges, we propose HoPE, a **H**ybrid **o**f **P**osition **E**mbedding designed to improve the long-context capability of VLMs. As illustrated in Figure 1 and Figure 2, HoPE first introduces a hybrid frequency allocation (HFA) strategy to better preserve the semantic preference property (Definition 3.1) in long-context modeling. Under this strategy, spatial information will be encoded with higher frequencies to capture local semantics, while the lowest frequencies will be set to zero (as in NoPE [30]) to facilitate long-range semantic modeling. Second, HoPE develops a dynamic temporal scaling (DTS) mechanism to enhance VLMs' robustness to various video speeds and enable flexible inference under diverse context lengths. We will detail these strategies as follows:

### 4.1 Hybrid Frequency Allocation Strategy

To extend vanilla RoPE to multimodal scenarios, a common approach is to allocate different frequencies to encode different positional components $(t, x, y)$. For example, M-RoPE [2] assigns the highest frequencies for temporal modeling and lower frequencies for spatial encoding. In contrast, VideoRoPE [28] allocates the lowest frequencies for temporal modeling, achieving better empirical results. However, in Theorem 3.1, we theoretically prove that, despite using lower frequencies being more ideal for semantic modeling, none of these frequency allocation strategies can maintain the ideal semantic preference property (Definition 3.1) over extended contexts.

To provide a stronger theoretical guarantee for the semantic preference property, we propose a hybrid frequency allocation strategy. As shown in Figure 1, we encode spatial information $(x, y)$ with high frequencies, as high frequencies are more sensitive to positional differences and thereby better at capturing local semantics [28, 38]. Following existing work [28], $x$ and $y$ are encoded in an interleaved manner to prevent biased spatial encoding. More importantly, unlike existing methods [23, 28, 2], we directly set the lowest frequencies to zero (as in NoPE [30]) to provide a stronger guarantee for the semantic preference property (Definition 3.1), as shown in Figure 2. Specifically, for $d = 128$, we interleave $x$ and $y$ positions in the first 96 dimensions of the rotation matrix and set the frequencies in the remaining 32 dimensions to zero, which corresponds to an identity matrix:

$$R_{\Delta x, \Delta y} = \text{diag}\left(\begin{pmatrix} \cos\theta_0\Delta x & -\sin\theta_0\Delta x & 0 & 0 & \cdots & 0 & 0 & 0 & 0 \\ \sin\theta_0\Delta x & \cos\theta_0\Delta x & 0 & 0 & \cdots & 0 & 0 & 0 & 0 \\ 0 & 0 & \cos\theta_1\Delta y & -\sin\theta_1\Delta y & \cdots & 0 & 0 & 0 & 0 \\ 0 & 0 & \sin\theta_1\Delta y & \cos\theta_1\Delta y & \cdots & 0 & 0 & 0 & 0 \\ \vdots & \vdots & \vdots & \vdots & \ddots & \vdots & \vdots & \vdots & \vdots \\ 0 & 0 & 0 & 0 & \cdots & \cos\theta_{46}\Delta x & -\sin\theta_{46}\Delta x & 0 & 0 \\ 0 & 0 & 0 & 0 & \cdots & \sin\theta_{46}\Delta x & \cos\theta_{46}\Delta x & 0 & 0 \\ 0 & 0 & 0 & 0 & \cdots & 0 & 0 & \cos\theta_{47}\Delta y & -\sin\theta_{47}\Delta y \\ 0 & 0 & 0 & 0 & \cdots & 0 & 0 & \sin\theta_{47}\Delta y & \cos\theta_{47}\Delta y \end{pmatrix}, I_{32}\right)$$

We now provide a theoretical analysis of how this hybrid strategy helps the attention mechanism to capture long-range semantic similarities. Building on Definition 3.1 and Theorem 3.1, we first formalize the condition under which semantic preference is preserved in multimodal RoPE.

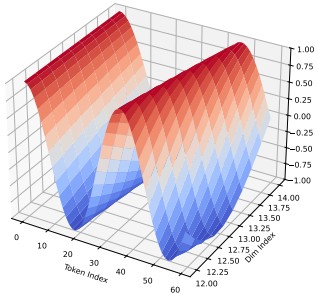 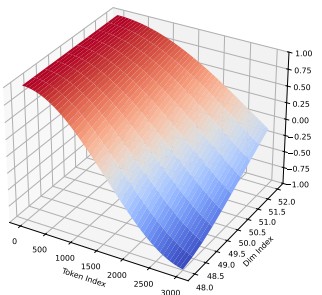 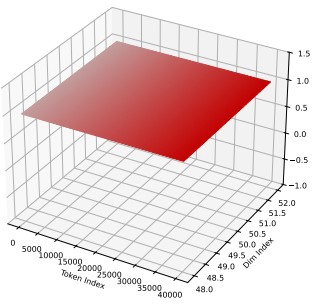

(a) High frequencies for temporal modeling in M-RoPE.

(b) Low frequencies for temporal modeling in VideoRoPE.

(c) Zero frequencies for temporal modeling in HoPE (ours).

Figure 2: **Multimodal RoPEs use different frequencies for temporal modeling.** M-RoPE uses the *highest* frequencies, which are suboptimal for long-context modeling. VideoRoPE utilizes the *lowest* frequencies for more stable semantic modeling. Our HoPE, employing *zero* frequencies for temporal modeling, establishes the upper bound of semantic modeling capabilities across all strategies.

In particular, Lemma 4.1 establishes a clear theoretical criterion for maintaining semantic preference with multimodal RoPE. It directly follows from our analysis in Theorem 3.1 and Appendix A.2, providing the theoretical foundation for our proposed method.

> **Lemma 4.1** (Necessary Condition for Semantic Preference). *For a multimodal RoPE with rotation matrix $\mathbf{R}_{t,x,y} = \mathrm{diag}(\mathbf{R}_t, \mathbf{R}_x, \mathbf{R}_y)$, the semantic preference property (Definition 3.1) holds if, for all possible relative distances,*
>
> $$\sum_{i \in i_t} 2\sigma^2 \cos(\Delta t \cdot \theta_i) + \sum_{i \in i_x} 2\sigma^2 \cos(\Delta x \cdot \theta_i) + \sum_{i \in i_y} 2\sigma^2 \cos(\Delta y \cdot \theta_i) \geq 0,$$
>
> *where $\sigma^2$ is the variance of each component in the query/key vector, $i_t, i_x, i_y$ are dimensions allocated to $t, x, y$, and $\Delta t \in \{0, 1, \ldots, L-1\}$, $\Delta x \in \{0, 1, \ldots, H\}$, $\Delta y \in \{0, 1, \ldots, W\}$.*

Based on this Lemma, we now prove how our hybrid frequency allocation strategy provides stronger guarantees for the semantic preference property over extended contexts. Specifically, HFA set $\theta_i = 0$ for all $i \in i_t$. Hence, the temporal terms in Lemma 4.1 reduce to $\sum_{i \in i_t} 2\sigma^2 \cdot 1$, noting that $\sum_{i \in i_t} 2\sigma^2 \cdot 1 \geq \sum_{i \in i_t} 2\sigma^2 \cos(\Delta t \cdot \theta_i)$ holds for any choice of temporal frequencies $\theta_i$. Adding the identical spatial terms on both sides, we obtain:

$$\begin{aligned}
& \sum_{i \in i_t} 2\sigma^2 \cdot 1 + \sum_{i \in i_x} 2\sigma^2 \cos(\Delta x \cdot \theta_i) + \sum_{i \in i_y} 2\sigma^2 \cos(\Delta y \cdot \theta_i) \\
& \geq \sum_{i \in i_t} 2\sigma^2 \cos(\Delta t \cdot \theta_i) + \sum_{i \in i_x} 2\sigma^2 \cos(\Delta x \cdot \theta_i) + \sum_{i \in i_y} 2\sigma^2 \cos(\Delta y \cdot \theta_i).
\end{aligned} \tag{4}$$

This shows that our HFA strategy, by setting the lowest frequencies to zero, dominates any other choice of temporal frequencies and provides a stronger guarantee for preserving the semantic preference property under long-context scenarios, as in Theorem 4.1.

**Theorem 4.1.** *For multimodal position embeddings with dimensions allocated across temporal ($i_t$), and spatial components ($i_x, i_y$), setting $\theta_i = 0$ for all temporal dimensions $i \in i_t$ maximizes the semantic preference guarantee in Definition 3.1, compared to any alternative frequency allocation strategy, particularly under extended context lengths.*

Another interesting finding is that, if we set $|i_t| = d/4, |i_x| = |i_y| = d/8$ and $\theta_i = 0, i \in i_t$, for any context length $t$ and spatial size $x, y$, semantic preference property invariably holds, as Lemma 4.1 reduces to $\sum_{i=0}^{d/8-1} 2\sigma^2(2 + \cos(\Delta x \cdot \theta_{2i}) + \cos(\Delta y \cdot \theta_{2i+1})) \geq 0$. However, the empirical results of this approach are inferior to our proposed HoPE, probably due to the decreased number of frequencies for spatial modeling. More discussions are provided in Appendix B.3.

## 4.2  Dynamic Temporal Scaling Mechanism

Considering the distinct information densities of text and visual tokens, HoPE introduces a dynamic temporal scaling mechanism that adjusts the temporal strides of visual inputs. Specifically, we first define a set of scaling factors, e.g., $\Gamma = \{0.5, 0.75, 1, 1.25, 1.5\}$, which includes both stretching ($\gamma > 1$) and compressing ($\gamma < 1$) operations. During training, the scaling factor $\gamma$ is randomly selected from $\Gamma$ and applied to each video. This allows the model to learn temporal relationships at multiple scales, making it more robust to variations in video speed, which are common in real-world scenarios. Consider a multimodal input $(text, video, text)$ of length $L_t$, $L_v$, and $L_e$, respectively. The position indices $(t, x, y)$ for each token with our dynamic scaling factor $\gamma$ are:

$$(t, x, y) = \begin{cases} (l, l, l), & 0 \le l < L_t \\[2mm] \begin{pmatrix} L_t + \gamma(l - L_t), \\ L_t + \gamma(l - L_t) + w - \frac{W}{2}, \\ L_t + \gamma(l - L_t) + h - \frac{H}{2} \end{pmatrix}, & L_t \le l < L_t + L_v \\[2mm] \begin{pmatrix} (\gamma - 1)L_v + l, \\ (\gamma - 1)L_v + l, \\ (\gamma - 1)L_v + l \end{pmatrix}, & L_t + L_v \le l < L_t + L_v + L_e \end{cases} \tag{5}$$

Note that for visual tokens ($L_t \le l < L_t + L_v$), $l - L_t$ indicates the distance of the current frame from the start frame. During inference, scaling factors can be flexibly selected from the set to accommodate videos of different lengths. It is worth noting that unlike existing methods, which do not consider temporal scaling for visual tokens [1, 2, 4, 5] or just apply a fixed and unidirectional scaling factor for both training and testing [28], our methods not only help the model learn temporal relationships at multiple scales, but also offer flexibility during inference to accommodate various context lengths.

## 5  Experiment

In this section, we evaluate the performance of HoPE on four video benchmarks across long video understanding and long video retrieval tasks, aiming to validate its effectiveness in multimodal long-context modeling. Additionally, we conduct ablation studies to investigate the individual contribution of each strategy to overall performance and the interplay between task type, context length, and scaling factor selection.

### 5.1  Experimental Setups

**Implementation Details.** We utilize Qwen2-1.5B and Qwen2-7B [39] as the backbone models. By integrating these models with vision encoders from Qwen2-VL-2B/7B-Instruct [2], we obtain Qwen2-2B/7B-Video, respectively. During training, we adopt a batch size of 128, a learning rate of 1e-5(2B)/2e-5(7B) with a cosine scheduler. Following the instruction tuning settings in Qwen2-VL [2], we set the maximum video frames to 128 and the video sampling rate to 2. The training context length is set to 8k, with the entire training process taking approximately 304 GPU hours on machines equipped with H800-80GB GPUs. During evaluation, the minimum tokens per frame are set to 144.

**Training Data.** We train the models on a subset of LLaVA-Video-178k [40], which consists of 178k videos ranging from 0 to 3 minutes and 5M instruction samples, including captions, free-form, and multiple-choice question answering. Our selected subset includes 30k videos with durations under 2 minutes and 3k videos with durations between 2 and 3 minutes, resulting in roughly 300k pairs.

**Baselines.** We compare HoPE with the following RoPE variants: 1) vanilla RoPE [23], the standard approach in long-context LLMs, 2) M-RoPE [2], a famous RoPE extension in Qwen2-VL for multimodal inputs, 3) VideoRoPE [28], a specialized RoPE variant designed for video-related tasks.

**Evaluation Benchmarks.** We evaluate HoPE across four video benchmarks for long video understanding and long video retrieval tasks. For long video understanding, we utilize LongVideoBench [41], Video-MME [42], and MLVU [43], covering videos ranging from a few seconds to 2 hours. For long video retrieval, we employ V-NIAH (Visual Needle-In-A-Haystack) [17]. In this task, a

Table 1: **Performance comparison on long video understanding benchmarks.** The training context length of all methods is set to 8k, and we report the performance on 8k, 16k, 32k, and 64k to evaluate length generalization. The best results are **bold**, while the second best results are underlined.

| Method | MLVU | | | | LongVideoBench | | | | Video-MME | | | |
|---|---|---|---|---|---|---|---|---|---|---|---|---|
| | 8k | 16k | 32k | 64k | 8k | 16k | 32k | 64k | 8k | 16k | 32k | 64k |
| *Qwen2-2B-Video* | | | | | | | | | | | | |
| Vanilla RoPE | **55.10** | 55.21 | 54.36 | 39.06 | 51.57 | 50.29 | 51.00 | 34.21 | 50.70 | 51.48 | 51.44 | 20.31 |
| M-RoPE | 53.26 | 53.69 | 54.73 | 40.63 | 50.81 | 52.26 | 51.30 | 44.74 | 51.44 | 51.22 | 51.52 | 23.44 |
| VideoRoPE | 54.75 | 55.19 | 54.00 | 42.19 | 52.17 | 52.02 | 51.31 | 36.84 | 50.89 | 50.52 | 50.56 | 15.63 |
| **HoPE (Ours)** | 54.89 | **56.36** | **55.70** | **45.12** | **52.31** | **52.97** | **51.66** | **46.27** | **51.79** | **51.87** | **51.69** | **26.03** |
| *Qwen2-7B-Video* | | | | | | | | | | | | |
| Vanilla RoPE | 59.75 | 61.13 | 61.03 | 34.38 | 51.17 | 50.31 | 51.29 | 39.47 | 56.70 | 57.96 | 57.99 | 26.13 |
| M-RoPE | 59.70 | 61.68 | 62.46 | 46.88 | 52.27 | 53.29 | 53.49 | 50.00 | 56.81 | 57.77 | 58.37 | 23.43 |
| VideoRoPE | 60.40 | 61.82 | 62.51 | 45.31 | 52.89 | 53.13 | 53.82 | 47.37 | 57.51 | 59.00 | 59.13 | 26.52 |
| **HoPE (Ours)** | **61.09** | **63.48** | **63.85** | **50.01** | **54.11** | **55.09** | **55.34** | **51.22** | **57.74** | **59.33** | **59.44** | **27.34** |

"needle" image is randomly inserted into a "haystack" video, and the VLM is required to answer a question specifically about the embedded "needle" image. Following the protocol in V-NIAH [17], we utilize a haystack video with 1-hour duration (3,000 frames) and insert the needle image at 20% depth intervals (e.g., a frame depth of 0% would place the needle image at the very beginning of the video). For more detailed benchmark descriptions, please refer to Appendix B.1.

## 5.2 Results on Long Video Understanding

In this section, we provide a comprehensive comparison of HoPE and different RoPE variants in long video understanding. From Table 1, we observe that: **(1)** HoPE consistently outperforms all baselines across nearly all benchmarks, context lengths, and backbone sizes. Specifically, under the 7B model scale and 32k context lengths, HoPE surpasses vanilla RoPE by 2.82, 4.05, and 1.45 on MLVU, LongVideoBench, and Video-MME, respectively. This confirms its effectiveness and generalizability in multimodal long-context modeling. **(2)** The effectiveness of HoPE scales with backbone size. For instance, when the size of the backbone LLM increases from 2B to 7B, HoPE's performance gain on LongVideoBench (32k) significantly increases from 0.66 to 4.05 compared to vanilla RoPE. Notably, the performance gap between different methods on the 2B scale is less significant, probably due to the limited capabilities of the backbone LLM. **(3)** For context lengths under 64k, performance on Video-MME drops substantially, while the impact on MLVU and LongVideoBench is less pronounced. This suggests that extrapolating to extreme context lengths (e.g., up to 8x) remains highly challenging.

## 5.3 Results on Long Video Retrieval

We evaluate HoPE against other RoPE variants on V-NIAH [17] to demonstrate the superiority of our method in long video retrieval, where VLMs are required to identify specific frames in a video to answer the question. Figure 3 demonstrates that multimodal RoPEs significantly outperform vanilla RoPE, supporting our claim in Proposition 3.1 that the flattening operation in vanilla RoPE hinders spatial-temporal modeling. Furthermore, HoPE achieves better extrapolation than M-RoPE and VideoRoPE, confirming its effectiveness in multimodal long-context modeling. Quantitative results in Table 4 show that HoPE surpasses the best baseline by a significant margin of 22.23%.

## 5.4 Analysis

In this section, we first conduct ablation studies to analyze the effectiveness of each component in HoPE. We then present a comprehensive analysis exploring how different factors, including task type, context length, and the scaling factor of visual tokens, interact and impact model performance.

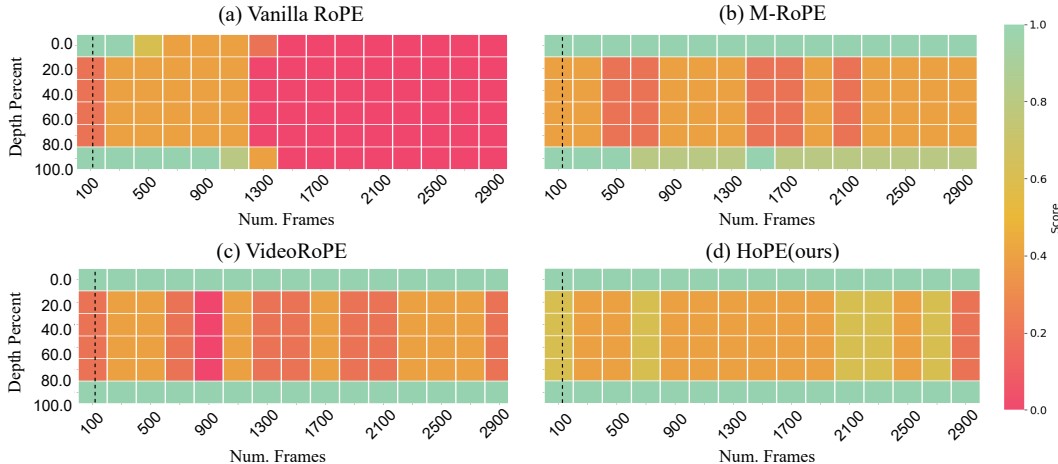

Figure 3: **Performance comparison on long video retrieval task (V-NIAH).** Here, each frame corresponds to 144 tokens. Cell colors indicate model accuracy (red: low, green: high), and the black dotted line marks the training context length (8k).

**Ablation Studies.** We conduct a series of ablation experiments to evaluate the impact of each component in HoPE and summarize the results in Table 2. According to the results, we observe that :

(1) The 3D structure effectively improves the performance of vanilla RoPE in multimodal contexts, supporting our Proposition 3.1. (2) Based on the 3D structure, the hybrid frequency allocation (HFA) strategy further enhances long-range semantic modeling, achieving an average improvement of 1.69 across all context lengths. (3) The dynamic temporal scaling (DTS) mechanism facilitates VLMs' robustness to varying video speeds in real-world scenarios, yielding further performance gain. By combining the above strategies, our HoPE achieves the best overall performance across different context lengths in multimodal long-context modeling.

Table 2: **Ablation results on Video-MME from 8k to 64k.** Here, HFA: hybrid frequency allocation, DTS: dynamic temporal scaling.

| Method | 8k | 16k | 32k | 64k |
|---|---|---|---|---|
| Vanilla RoPE | 56.70 | 57.96 | 57.99 | 26.13 |
| + 3D structure | 56.81 | 57.77 | 58.37 | 23.43 |
| + 3D + HFA | 57.66 | 59.19 | 59.31 | 26.98 |
| + 3D + HFA + DTS | **57.74** | **59.33** | **59.44** | **27.34** |

**Impact of Test-time Scaling Factor Selection.** We conduct further experiments to investigate how different scaling factors $\gamma$ in our dynamic temporal scaling mechanism impact the performance of video-related tasks. We summarize the results on V-NIAH and LongVideoBench in Table 3. Our main observations are as follows: **(1)** *Long video retrieval generally prefers smaller scaling factors.* As shown in Table 3, when we utilize smaller scaling factors $\gamma$ during inference, the performance on V-NIAH improves. We attribute this to the substantial length of 1-hour videos

Table 3: **Ablation studies on test-time scaling factor selection.** We find that long video understanding generally benefits from larger scaling factors, while long video retrieval yields better results with smaller ones.

| Scaling Factor $\gamma$ | LongVideoBench | | | | V-NIAH |
|---|---|---|---|---|---|
| | 8k | 16k | 32k | 64k | |
| 0.50 | **54.48** | 54.29 | 54.36 | 52.63 | 60.89 |
| 0.75 | 54.36 | 54.97 | 54.72 | 52.63 | **63.56** |
| 1.00 | 54.11 | 54.48 | 54.97 | 52.63 | 62.67 |
| 1.25 | 54.11 | 54.84 | **55.70** | 52.63 | 62.67 |
| 1.50 | 54.11 | **55.09** | 55.34 | 51.22 | 61.78 |

(3,000 frames), which far exceeds the training length (128 frames). In such cases, smaller scaling factors indirectly prevent the spatial position indices from becoming excessively large (see Equation 5), thereby providing a better guarantee for the semantic preference property. Therefore, we set $\gamma = 0.75$ for long video retrieval. **(2)** *Long video understanding generally benefits from larger scaling factors.* In contrast to retrieval, we find that long video understanding is relatively insensitive to the choice of scaling factor when the input context length is close to the training length. However, as the input length increases, employing larger scaling factors ($\gamma > 1$) results in better performance. We hypothesize that while smaller scaling factors help preserve the semantic preference property, larger

scaling factors are beneficial for maintaining spatial details (also see Equation 5), which are crucial for complex understanding tasks. This introduces a natural tradeoff between semantic preference and spatial detail preservation. Compared to long video retrieval (3,000 frames, roughly 432k tokens), where extended temporal distances can significantly degrade semantic preference, in long video understanding tasks with context lengths of 16k–32k, the negative impact on semantic preference is relatively small. At the same time, the positive effect of larger scaling factors on capturing spatial details outweighs the semantic preference loss, making larger scaling factors overall more effective for complex video understanding. In our experiments, we set $\gamma = 1.5$ for long video understanding.

# 6  Related Work

**Position Embedding in LLMs.** Rotary Position Embedding (RoPE) [23] has become a common choice for position embedding in modern LLMs [24–26, 44]. As discussed in Section 2, RoPE achieves this success through rotating query and key vectors, encoding *relative* position information through an *absolute* positional encoding approach. Despite its success, several works have pointed out that No Position Embedding (NoPE) still works for decoder-only LLMs, arguing that the causal attention mechanism implicitly learns *absolute* position information [30, 31, 38]. These works even suggest that NoPE outperforms RoPE in out-of-distribution (OOD) scenarios. However, this observation remains unexplored in multimodal settings, where positional encoding strategies may have different implications for cross-modal interactions. Based on Lemma 4.1, we find that incorporating NoPE's zero frequency strategy indeed improves the length generalization of multimodal RoPE.

**Multimodal Position Embedding in VLMs.** In VLMs [1–5], images are first processed by vision encoders and then flattened into 1D tokens. Several early models [1, 4, 5] rely on vanilla RoPE for positional encoding, which distorts spatial-temporal locality (see Section 3) and limits VLMs' long-context capability. Recently, Qwen2-VL [2] introduced M-RoPE, which extends 1D RoPE to multimodal settings by assigning distinct frequency ranges to different positional components. Specifically, M-RoPE allocates the *highest* frequencies to the temporal component $t$, while distributing the lower frequencies sequentially to the spatial components $x$ and $y$. Conversely, VideoRoPE [28] allocates the *lowest* frequencies to $t$ to capture long-range dependencies, achieving stronger length generalization. However, these allocation strategies mainly rely on heuristics, lacking in-depth theoretical analysis. In contrast, our work theoretically analyzes how different frequency allocation strategies impact the performance of multimodal RoPE. By zeroing out low frequencies for temporal modeling, our proposed HoPE provides the strongest theoretical guarantee for long-range semantic modeling. HoPE's strength is further enhanced by its dynamic temporal scaling of visual tokens, which enables robust temporal learning during training and flexible scaling during inference. By integrating these advantages, HoPE achieves state-of-the-art performance in long video understanding and retrieval tasks, making it well-suited for long context VLMs.

# 7  Conclusion

This paper theoretically analyzes the limitations of current multimodal RoPE variants. Our analysis reveals that: (1) vanilla RoPE inherently fails in spatial-temporal modeling; (2) keeping all frequencies in multimodal RoPE is unreliable in capturing long-range semantic similarities; (3) temporal scaling of lengthy visual tokens should include both compression and expansion to accommodate various video speeds. Consequently, we introduce HoPE, a hybrid of position embedding designed to enhance the long-context capabilities of VLMs. HoPE proposes a hybrid frequency allocation strategy to facilitate long-range semantic modeling, and a dynamic temporal scaling mechanism to enhance VLMs' robustness to varying video speeds in real-world scenarios. Experimental results on long video understanding and long video retrieval tasks demonstrate that HoPE consistently outperforms existing methods across diverse context lengths and backbone sizes, confirming its effectiveness.

**Limitations.** While HoPE's performance gains scale from 2B to 7B backbones, our work does not use larger models or training data. We observe that the performance of all methods degrades significantly at 64k, though HoPE remains the most robust. While these resource-constrained evaluations are essential for uncovering genuine algorithmic benefits of multimodal RoPE, we note that training with more data, particularly long-context data, could further improve length generalization.

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

# A  Proofs

In this section, we provide detailed proofs for the theoretical statements presented in this paper.

## A.1  Vanilla RoPE Fails in Spatial-Temporal Structure

**Proposition 3.1.** *Given any query $\mathbf{q}$ at position $(t, x, y)$ and a relative distance of 1 in spatial or temporal dimensions, the flattening operation in 1D RoPE distorts the relative distance with a magnitude dependent on the frame resolution.*

*Proof.* Consider a video of shape $T \times H \times W$, where each token at position $(t, x, y)$ is flattened by

$$f(t, x, y) = tHW + xW + y.$$

Now consider two types of local neighbors:

1. **Spatial neighbors within the same frame**:

Let $(t, x, y)$ and $(t, x + 1, y)$ be adjacent in the spatial dimension. Then,

$$|f(t, x + 1, y) - f(t, x, y)| = |((x+1)W + y) - (xW + y)| = W. \tag{6}$$

Note that a relative distance of 1 in $x$ becomes $W$ after flattening, which grows linearly with the frame width.

2. **Temporal neighbors at the same spatial position**: Let $(t, x, y)$ and $(t + 1, x, y)$ be adjacent in time. Then,

$$|f(t + 1, x, y) - f(t, x, y)| = |(t+1)HW + xW + y - (tHW + xW + y)| = HW. \tag{7}$$

For a 1-frame shift in time, the index difference becomes $HW$, which grows with spatial resolution.

In both cases, spatially or temporally adjacent tokens are mapped to indices with significant differences. Since vanilla RoPE incorporates positional information based on these 1D index differences, such flattening leads to distorted spatial-temporal relationships. □

## A.2  Semantic Preference Property

We now prove that the frequency allocation strategies in current multimodal RoPEs are unreliable in capturing semantic similarities over extended contexts, as defined in Definition 3.1.

**Definition 3.1.** (Semantic Preference). For any query vector $\mathbf{q}$ and a semantically similar key vector $\mathbf{k}'$ that can be expressed as $\mathbf{k}' = \mathbf{q} + \delta$ where $\delta$ is a zero-mean perturbation, the attention score with RoPE should satisfy:

$$\mathbb{E}_{\mathbf{q}, \mathbf{k}, \delta}[\mathbf{q}\mathbf{R}_{\Delta t \Delta x \Delta y}\mathbf{k}' - \mathbf{q}\mathbf{R}_{\Delta t \Delta x \Delta y}\mathbf{k}] \geq 0, \tag{8}$$

where $\mathbf{k}$ is the key vector of a semantically unrelated token. This preference should hold regardless of the relative distance $(\Delta t, \Delta x, \Delta y)$ between query-key pairs.

Firstly, we use Lemma A.1 to show why using lower frequencies for temporal modeling is more ideal in multimodal RoPE. Intuitively, larger rotation angles (frequencies) are more likely to produce negative cosine similarity values between semantically related tokens under long-context scenarios.

**Lemma A.1.** *Let $\Delta t$ be drawn uniformly from $\{0, 1, \ldots, L - 1\}$, and define*

$$P_\neg(\theta) = \frac{1}{L} \left| \{\Delta : \cos(\theta \, \Delta t) < 0\} \right|.$$

*Then for any $L > 1$:*

1. *If $0 < \theta < \frac{\pi}{2(L-1)}$, then $P_\neg(\theta) = 0$.*

2. *For $\theta \geq \frac{\pi}{2(L-1)}$, $P_\neg(\theta)$ is non-decreasing in $\theta$.*

3. *$\lim_{\theta \to \infty} P_\neg(\theta) = \frac{1}{2}$.*

*Proof of Lemma A.1.* 1. **No negative region for small $\theta$.** If $0 < \theta < \frac{\pi}{2(L-1)}$, then for every $\Delta t \in \{0, \ldots, L-1\}$ we have

$$0 \;\leq\; \theta \, \Delta t \;<\; \theta \, (L-1) \;<\; \pi/2,$$

so $\cos(\theta \, \Delta t) > 0$. Hence $P_\neg(\theta) = 0$.

2. **Monotonicity once the first zero enters.** As soon as $\theta \geq \frac{\pi}{2(L-1)}$, the point satisfying $\theta \, \Delta t = \pi/2$ lies in $\{0, \ldots, L-1\}$. Each further increase in $\theta$ extends the interval of length $\theta(L-1)$, adding more half-periods of cosine. Each added half-period contains exactly one "negative" region of length $\pi$. Therefore the count $\left| \{\Delta : \cos(\theta \Delta) < 0\} \right|$ (and hence $P_\neg(\theta)$) can only stay the same or increase, up to $O(1/L)$ rounding errors on the discrete grid.

3. **Limit to one half for large $\theta$.** For large $\theta$, the values $\{\theta \Delta\}_{\Delta=0}^{L-1}$ become equidistributed mod $2\pi$. Since the negative region $\{\, x \mod 2\pi \mid \cos x < 0 \,\}$ has total length $\pi$ over each $2\pi$-cycle, one finds

$$\lim_{\theta \to \infty} P_\neg(\theta) \;=\; \frac{\pi}{2\pi} \;=\; \tfrac{1}{2}.$$

$\square$

We can now prove the frequency allocation strategies in current multimodal RoPE cannot reliably maintain the semantic preference property, i.e., semantically similar tokens should receive higher attention than semantically unrelated pairs.

**Theorem 3.1.** *Let $X = [x_1, x_2, \ldots, x_L]$ be an input sequence, and let RoPE use any fixed set of temporal frequencies (e.g., highest or lowest). Then there exists a critical length $L_c$ such that for all $L \geq L_c$, the semantic preference property (Definition 3.1) is violated.*

*Proof.* We first recall the definition of multimodal RoPE, where the rotation matrix is partitioned to encode different dimensions:

$$\mathbf{R}_{t,x,y} = \mathrm{diag}(\mathbf{R}_t, \mathbf{R}_x, \mathbf{R}_y),$$

where $\mathbf{R}_t$, $\mathbf{R}_x$, and $\mathbf{R}_y$ are rotation matrices applied to temporal, horizontal spatial, and vertical spatial dimensions, respectively, with each dimension carrying a frequency of $\theta_i = b^{-2i/d}, i \in \{0, \ldots, d/2 - 1\}$. Note that the $(t, x, y)$ ordering is purely notational and does not constrain the actual dimension allocation strategy.

Assume that each component of the query vector $\mathbf{q}$ is independently and identically distributed with mean $\mu$ and variance $\sigma^2$. We denote key vector that is semantically similar to $\mathbf{q}$ as $\mathbf{k}' = \mathbf{q} + \delta$, where $\delta$ is a zero-mean perturbation. The semantically unrelated key vector $\mathbf{k}$ is independently drawn with the same distribution as $\mathbf{q}$. Let $\Delta t, \Delta x, \Delta y$ denote relative temporal and spatial distances between the query and each key. According to Definition 3.1, the semantic preference property requires that:

$$\mathbb{E}_{\mathbf{q},\mathbf{k},\delta}[\mathbf{q}\mathbf{R}_{\Delta t, \Delta x, \Delta y}\mathbf{k}'^\top - \mathbf{q}\mathbf{R}_{\Delta t, \Delta x, \Delta y}\mathbf{k}^\top]$$

$$= \mathbb{E}_{\mathbf{q},\mathbf{k},\delta}[\mathbf{q}\mathbf{R}_{\Delta t, \Delta x, \Delta y}(\mathbf{q} + \delta)^\top - \mathbf{q}\mathbf{R}_{\Delta t, \Delta x, \Delta y}\mathbf{k}^\top]$$

$$= \mathbb{E}_{\mathbf{q}}[\mathbf{q}\mathbf{R}_{\Delta t, \Delta x, \Delta y}\mathbf{q}^\top] - \mathbb{E}_{\mathbf{q},\mathbf{k}}[\mathbf{q}\mathbf{R}_{\Delta t, \Delta x, \Delta y}\mathbf{k}^\top]$$

$$= \mathbb{E}_{\mathbf{q}}[\mathbf{q}\mathbf{R}_{\Delta t, \Delta x, \Delta y}\mathbf{q}^\top] - \mu^2 \mathbf{R}_{\Delta t, \Delta x, \Delta y}$$

$$= \sum_{i \in i_t} 2(\mu^2 + \sigma^2)\cos(\Delta t)\theta_i + \sum_{i \in i_x} 2(\mu^2 + \sigma^2)\cos(\Delta x)\theta_i + \sum_{i \in i_y} 2(\mu^2 + \sigma^2)\cos(\Delta y)\theta_i - \quad (9)$$

$$\sum_{i \in i_t} 2\mu^2\cos(\Delta t)\theta_i + \sum_{i \in i_x} 2\mu^2\cos(\Delta x)\theta_i + \sum_{i \in i_y} 2\mu^2\cos(\Delta y)\theta_i$$

$$= \sum_{i \in i_t} 2\sigma^2\cos(\Delta t \cdot \theta_i) + \sum_{i \in i_x} 2\sigma^2\cos(\Delta x \cdot \theta_i) + \sum_{i \in i_y} 2\sigma^2\cos(\Delta y \cdot \theta_i) \geq 0,$$

where $i_t, i_x, i_y$ denote dimensions allocated to encode temporal $(t)$, horizontal spatial $(x)$, and vertical spatial $(y)$ information. To satisfy the semantic preference property (Definition 3.1), the expected attention between a query and its semantically similar key should remain higher than that

for an unrelated key, regardless of their relative distance. This implies the following condition must hold universally:

$$\sum_{i \in i_t} 2\sigma^2 \cos(\Delta t \cdot \theta_i) + \sum_{i \in i_x} 2\sigma^2 \cos(\Delta x \cdot \theta_i) + \sum_{i \in i_y} 2\sigma^2 \cos(\Delta y \cdot \theta_i) \geq 0,$$
$$\Delta t \in \{0, 1, \ldots, L-1\}, \Delta x \in \{0, 1, \ldots, H\}, \Delta y \in \{0, 1, \ldots, W\}. \tag{10}$$

Now consider a long-context scenario, where $L \gg H, W$, we can now theoretically prove that why VideoRoPE [28] (using lowest frequencies for $t$) is better than M-RoPE (using highest frequencies for $t$) in maintaining semantic preference property in long contexts. Simply, by Lemma A.1, we show that when the context length $L$ is sufficiently large, the probability that $cos(\Delta t \cdot \theta_i)$ leads to negative values becomes higher when $\theta_i$ becomes larger. Therefore, lower frequencies, which rotate less, are less likely to violate the semantic preference property.

However, despite using the lowest frequencies, VideoRoPE still fails to guarantee that the semantic preference property holds for all context lengths (Equation 10). Let VideoRoPE allocate only the smallest frequency to the temporal dimensions, instead of $|i_t|$ smallest frequencies:

$$\theta_{\min} = b^{-2(\frac{d}{2}-1)/d},$$

so that in Equation (10) the temporal sum reduces to:

$$2\sigma^2 |i_t| \cos(\Delta t \cdot \theta_{\min}).$$

Here, under the reasonable assumption that semantically related tokens co-occur in nearby spatial positions across frames, the spatial sums in Equation 10 remains non-negative. Thus we only consider the temporal sum in Equation 10:

$$2\sigma^2 |i_t| \cos(\Delta t \cdot \theta_{\min}).$$

Now pick any context length $L$ so large that there exists

$$\Delta t \in \{0, 1, \ldots, L-1\} \quad \text{with} \quad \Delta t\, \theta_{\min} \in \left(\frac{\pi}{2}, \frac{3\pi}{2}\right).$$

Such a $\Delta t$ indeed exists as soon as $\theta_{\min}(L-1) > \frac{\pi}{2}$, i.e. for any

$$L > L_c = \frac{\pi}{2\,\theta_{\min}} + 1.$$

For that choice of $\Delta t$, we have

$$\cos(\Delta t \cdot \theta_{\min}) < 0,$$

and hence the left-hand side of Equation (10) becomes

$$2\sigma^2 |i_t| \cos(\Delta t \cdot \theta_{\min}) < 0.$$

This single counterexample $(\Delta t, \Delta x, \Delta y)$ violates the semantic preference condition, since no further temporal frequencies are available to "rescue" the sum. Therefore, despite using the lowest frequencies for temporal modeling, VideoRoPE still fails to guarantee the semantic preference property. In conclusion, all frequency allocation strategies in current multimodal RoPEs fail to maintain the semantic preference property in Definition 3.1, completing the proof. $\square$

## B  Further Experimental Details

In this section, we provide further details of our experiments, including benchmark descriptions, experimental settings, and further results.

### B.1  Detailed Benchmark Description

There is a growing interest in video generation and understanding [45–47, 34, 19], given their broad applications in content creation and analysis. In this subsection, we provide detailed descriptions of the video benchmarks we used in the experiments, i.e., LongVideoBench [41], Video-MME [42], MLVU [43], and V-NIAH [17].

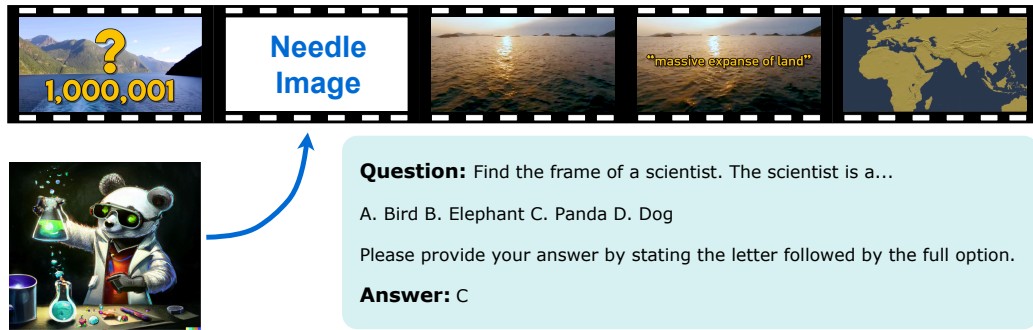

Figure 4: Illustration of V-NIAH, which consists of a randomly inserted needle image, a haystack video, and a specific question related to the needle.

- **LongVideoBench** is a comprehensive benchmark for evaluating Vision-Language Models on long video understanding tasks. Unlike traditional video benchmarks that focus on short clips under one minute, this dataset features videos ranging from 8 seconds to 1 hour across diverse sources, including everyday life, movies, knowledge, and news. The benchmark encompasses 17 fine-grained question categories organized into two levels: perception and relation. In our experiment, questions that are free from subtitles are retained.

- **Video-MME** is a full-spectrum evaluation benchmark of Vision-Language Models in video analysis, spanning 6 primary visual domains with 30 subfields to ensure generalizability. It features temporal diversity by incorporating both short- (<2 minutes), medium- (4-15 minutes), and long-term videos (30-60 minutes), ranging from 11 seconds to 1 hour.

- **MLVU** is a high-quality benchmark designed to evaluate the video understanding capabilities of Vision-Language Models. The temporal duration of videos within MLVU spans from 3 minutes to 2 hours, covering genres such as movies, life records, and egocentric videos. In our experiment, we evaluate all methods on the following multiple-choice tasks: Action Count, Action Order, Topic Reasoning, Ego Reasoning, Needle QA, Plot QA, and Anomaly Recognition.

- **V-NIAH** is a challenging benchmark designed to evaluate VLMs' ability to identify specific frames within long videos. In this task, a "needle" image is inserted into a "haystack" video, and the VLMs are required to answer specific questions about this "needle" image, as shown in Figure 4. Following the settings in V-NIAH [17], we utilize a haystack video with 1-hour duration (3,000 frames). The needle image is inserted at 20% depth intervals (e.g., a frame depth of 0% would place the needle image at the very beginning of the video.)

Table 4: Quantitative performance of different RoPE variants on V-NIAH. Here, we report the average accuracy across different context lengths and frame depths.

|  | Vanilla RoPE | M-RoPE | VideoRoPE | HoPE (ours) |
| --- | --- | --- | --- | --- |
| V-NIAH | 21.00 | 47.11 | 52.00 | **63.56** |

## B.2 Quantitative Results on V-NIAH

Here, we provide the quantitative results of different RoPE variants on long video retrieval task in Table 4. It can be observed that our HoPE demonstrates a 22.23% improvement compared to the best baseline, justifying its effectiveness in multimodal long-context modeling.

## B.3 Ideal Condition for Semantic Preference

As discussed after Theorem 4.1, the semantic preference property (Definition 3.1) invariably holds for any context length $t$ and spatial size $x, y$ when we set $|i_t| = d/4, |i_x| = |i_y| = d/8$ and $\theta_i = 0, i \in i_t$, since Lemma 4.1 reduces to:

$$\sum_{i=0}^{d/8-1} 2\sigma^2(2 + \cos(\Delta x \cdot \theta_{2i}) + \cos(\Delta y \cdot \theta_{2i+1})) \geq 0.$$

In our original HoPE implementation, the frequencies allocated to $t, x, y$ are $16, 24, 24$, respectively, with the lowest 16 frequencies for $t$ set to zero. For this proposed variant (HoPE-X), we redistribute these allocations to $32, 16, 16$ for $t, x, y$, respectively, while setting the lowest 32 frequencies for $t$ to zero. To evaluate the comparative effectiveness of these configurations, we conduct further experiments on LongVideoBench.

Table 5: Performance comparison between HoPE-X and HoPE.

| Method | LongVideoBench | | | |
|--------|------|------|------|------|
|        | 8k   | 16k  | 32k  | 64k  |
| HoPE-X | 52.68 | 52.73 | 53.01 | 46.32 |
| HoPE   | 54.11 | 55.09 | 55.34 | 51.22 |

Table 5 demonstrates that HoPE consistently outperforms HoPE-X across diverse context lengths. We deduce that the inferior performance of HoPE-X is due to its decreased dimensions allocated for spatial modeling. While this configuration helps to maintain the semantic preference property, it negatively impacts HoPE-X's ability to model local features. Therefore, it is necessary to keep adequate dimensions for spatial modeling in multimodal RoPE.

