# OpenReview forum: "HoPE: Hybrid of Position Embedding for Long Context Vision-Language Models"
_NeurIPS.cc/2025/Conference — NeurIPS 2025 poster_

### Official Review · Reviewer_iTBi · 2025-06-26

**Clarity:** 2
**Significance:** 2
**Originality:** 3
**Rating:** 4
**Confidence:** 4

**Summary:**

This paper focuses on improving the RoPE for long context generalization in VLMs. The authors first analyse existing RoPEs that fail to generalize to long contexts. Then, based on the analysis, the authors propose a novel hybrid RoPE, trying to solve the generalization challenge. Extensive experiments are conducted to evaluate the proposed method.

**Questions:**

NA.

**Ethical Concerns:**

["NO or VERY MINOR ethics concerns only"]

**Final Justification:**

Thanks the authors for the reply. I am not fully convinced.

From the results on 8k - 32k, I find the evidence that HoPE performs better than previous baseline works, even though the experimental setting and results are quite different from the baseline work Video-RoPE.  I will raise my score.

However, results on 64k are still a major concern. The long-context generalization ability of HoPE is not fully verified on the 64k setting. Video MME 64k acc is only 27.34%, only slightly better than a random guess. This cannot fully support the claim of solving the long-context generalization issue and outperforming Video-RoPE.

The theoretical analysis is really interesting and I encourage the authors to align the experiment settings and the results with baseline works like Video-RoPE, which could be more convincing.

**Limitations:**

yes

**Quality:**

3

**Strengths And Weaknesses:**

Strengths:
1. Theoretical analysis is provided for both the existing RoPE and the proposed RoPE.
2. The proposed HoPE is interesting and novel.

Weaknesses:
1. Experiments cannot support the claim. We can see from Table 1, when scaling up from 8k to 64k sequence length, performance drops significantly and consistently across various benchmarks. The generalization problem is not solved at all to me.
2. I checked the baseline method, VideoRoPE, which was mentioned and compared in this paper. In their paper Table 2., VideoRoPE does not have performance drops (even slightly increasing, which totally makes sense) when scaling the context sequence length.
3. Potential wrong/unfair evaluation.  I do not know why in this paper, the reported VideoRoPE results drop significantly from 8k to 64k.

VideoRoPE, https://arxiv.org/pdf/2502.05173

Given the above major concerns, I think this paper is far below neurips standard, even if the theoretical analysis seems interesting.

---

> ### Author Rebuttal · Authors · 2025-07-28
>
> Thanks for your review. We respectfully disagree with your current comments and assessment of our work.
>
> Overall, the three weaknesses mentioned here point to only one opinion held by the reviewer: because our 64k context length results seem to be different from those in VideoRoPE [1], the reviewer directly concludes that our evaluation is wrong, and even our paper is far below the NeurIPS standard. We would like to point out that:
>
> - **Difference in training data:** The training data used by VideoRoPE [1] and our paper is different. Please refer to Section 5.1 of VideoRoPE and Section 5.1 of our paper. Given this fact, it is unfair to assume that our experiments must yield the same results/trends as VideoRoPE [1].
> - **Existing evidence supporting our observations:** The reviewer holds the view that when increasing the input context length, the performance should increase as reported in VideoRoPE [1]; otherwise, it doesn't make sense on their side. However, the reviewer might not be familiar with current works on long-context VLMs. In Table 5 of LongVideoBench's paper [2], we can observe that VLMs (Idefics2 [3] and Mantis-Idefics2 [4]) with limited capabilities would face performance degradation when the input context is longer. Given the fact that all methods are trained from scratch in our paper with less compute and data compared to powerful open/closed-source models, it is expected that performance degrades under context length (64k) that are much longer than the training length (8k). Moreover, it can be observed that it is **not only** VideoRoPE [1], but all methods in our paper face performance degradation under 64k, with our HoPE being the most robust.
>
> Given the above facts, it would be a rigid, unsupported expectation that all results must mirror those in prior work (particularly VideoRoPE [1] mentioned by the reviewer here), despite clear experimental differences.
>
> We would appreciate it if the reviewer would reevaluate our work.
>
> ***
> [1] Wei et al. VideoRoPE: What Makes for Good Video Rotary Position Embedding? ICML 2025.
>
> [2] Wu et al. Longvideobench: A Benchmark for Long-Context Interleaved Video-Language Understanding. NeurIPS 2024.
>
> [3] Laurençon et al. What Matters When Building Vision-Language Models? NeurIPS 2024.
>
> [4] Jiang et al. Mantis: Interleaved Multi-Image Instruction Tuning. TMLR 2024.

---

> ### Comment · Reviewer_iTBi · 2025-08-05
> **Discussion**
>
> Thanks the authors for the reply. However, I am not convinced.
>
> 1. Difference in training data is not an excuse. Both works use open-sourced data Llava-video-178k, with different random subsets. The author could easily use similar training data to train the model. Otherwise, on their customized training data, the model performance should be aligned with previous results. However, in this work, as I stated in the weakness, all the experiments show extremely different results from previous baselines, which makes the proposed method less convincing and casts doubt on the true effectiveness.
>
> 2. I am an expert in long-context VLMs and have experience with continuous pretraining Qwen-VL from 32k to 128k (now even 256k). We did not observe such a performance drop on M-RoPE. This is consistent with the reported results from Video-RoPE. Moreover, scaling up context length while causing a significant performance drop is not acceptable. Why would we need a worse model with a larger context window size?
> ##  Say in Table 1 of this work, HoPE on VideoMME 64k acc is only 27.34%, it seems just like a random guess from 4 options. There must be something wrong with the experiments.
>
> Overall, the key point is, the author claims to solve the context length generalization through the proposed HoPE, but the experiments show that HoPE will cause a significant performance drop when scaling up the sequence length from 8k to 64k, which is not acceptable and could not support the claim.
>
> The authors try to point out that other methods in their experiments also show performance degradation, and HoPE is better than other methods. However, the point here is, in baseline works like Video-RoPE, the original reported results do not have such a degradation. And based on my own experience, M-RoPE also does not have such a significant degradation. These cast doubt on the true effectiveness of the proposed method.
>
> Therefore, I still lean towards rejection.
>
> The theoretical analysis is really interesting and I encourage the authors to align the experiment settings and the results with baseline works like Video-RoPE, which could be more convincing.

---

> ### Author Response · Authors · 2025-08-05
>
> Thank you for your time to engage in the discussion after the PC's reminder. However, we kindly remind that an emotional and inconsistent review risks being less convincing and effective.
>
>
> > **Q1:** Difference in training data is not an excuse. Both works use open-sourced data Llava-video-178k, with different random subsets. The author could easily use similar training data to train the model. Otherwise, on their customized training data, the model performance should be aligned with previous results.
>
> A1: **Subjective assertions.** Beyond subjective assertions, it seems that the reviewer has failed to justify: (1) why our training data "must" align with VideoRoPE; (2) why differences in data composition, known to heavily influence model performance, should be dismissed as "no excuse." In fact, we are confused by the reviewer's assertion, "The author could easily use similar training data to train the model." Which part of the data do you refer to, and what could be "similar" in your definition? In reality, our data choices were driven by two simple facts: (1) with limited computation resources, we followed Mantis [1] and selected roughly 300k pairs for training (the size of primary multi-image dataset in Mantis [1]); and (2) at the time we conducted our work, neither VideoRoPE's data nor its peer review record were publicly available. Therefore, why would we "automatically" follow an unreleased setup?
> ***
>
> > **Q2:** However, in this work, as I stated in the weakness, all the experiments show extremely different results from previous baselines
>
> A2: **Inconsistent review.** We kindly remind the reviewer that in the weakness of your first review, you questioned only the 64k context length results. We are curious that you now claim that "all the experiments" are suspect. We are happy to discuss specific questions regarding results on other context lengths (e.g., 8k, 16k, 32k). Otherwise, emotional review and discussion risk being less effective.
>
>
> ***
>
> > **Q3:** I am an expert in long-context VLMs... We did not observe such a performance drop on M-RoPE. This is consistent with the reported results from Video-RoPE. Moreover, scaling up context length while causing a significant performance drop is not acceptable.
>
> A3: **Arbitrary claims.** We find it difficult to accept the reviewer's claim that VideoRoPE's results are unassailable "golden rule" despite clearly different setups. Overall, the whole point is solely based on the reviewer's assertion that he/she is an expert. We trust the AC, SAC, and PC will consider the full context. Likewise, calling a performance drop at longer contexts "not acceptable" without justification is confusing to us. Table 5 of LongVideoBench [2] shows that when increasing `max_frames` of input context, open-source VLMs (including Idefics2 [3] and Mantis-Idefics2 [1]) face significant performance degradation (~20 points), as they are trained with less compute and data (similar to ours). We have covered this point in our rebuttal, but the reviewer seems to miss this. If you dispute this, we are happy to discuss further.
>
> ***
>
> > **Q4:** Say in Table 1 of this work, Video MME 64k acc is only 27.34%, it seems just like a random guess from 4 options. There must be something wrong with the experiments.
>
> A4: Firstly, this significant degradation **only** happens at 64k on Video-MME, not including LongVideoBench and MLVU benchmarks. We attribute this to less long clips in our training data (*six times less*), which makes it more challenging for models to extrapolate from 8k pre-training length to 64k inference length. However, it would be ungrounded to assert that there **"must"** be something wrong with our experiments. If you have grounded suspicions, we are happy to discuss.
>
> ***
>
> > **Q5:** However, the point here is, in baseline works like Video-RoPE, the original reported results do not have such a degradation. And based on my own experience, M-RoPE also does not have such a significant degradation.
>
> A5: There exist clear experimental differences, as we stated in the first rebuttal. If the reviewer only bases his disputes on his impressive confidence in VideoRoPE and so-called experience, we are unable to continue meaningful discussion.
>
> ***
>
> Finally, we appreciate your acknowledgment of our theoretical analysis. However, we respectfully disagree with your assertion that relevant works "must" align the settings with the specific VideoRoPE; otherwise, they are not convicing at your side. It would be more convicing and effective to base discussions on objective evidence rather than assertions and claims.
>
> ***
>
> [1] Jiang et al. Mantis: Interleaved Multi-Image Instruction Tuning. TMLR 2024.
>
> [2] Wu et al. Longvideobench: A Benchmark for Long-Context Interleaved Video-Language Understanding. NeurIPS 2024.
>
> [3] Laurençon et al. What Matters When Building Vision-Language Models? NeurIPS 2024.

---

> > ### Comment · Reviewer_iTBi · 2025-08-05
> >
> > Thanks authors for the detailed reply.
> >
> > I think the key point we are discussing right now is whether one should align the experiment settings with baseline works or not.
> >
> > I totally understand that, solely based on the experimental result in this work's setting, HoPE is better than baseline works.
> >
> > But in my **personal** opinion, one qualified academic paper should conduct comparable experiments based on existing baseline works. Academic progress relies on the ability to isolate the impact of a new method by minimizing variables, especially when claiming to address a problem (long-context generalization) that prior work (e.g., VideoRoPE) has already tackled.

---

> > > ### Author Response · Authors · 2025-08-05
> > >
> > > Thank you for your response and for acknowledging the fairness and consistency of our experimental settings and results.
> > >
> > > Firstly, we think the key point we are discussing is not "whether one should align the experiment settings with baseline works or not." Instead, we think the primary argument raised by the reviewer here is "why our paper's results (particularly 64k results) are different from those in VideoRoPE, and whether this means wrong evaluation."
> > >
> > > In our rebuttal, we have comprehensively explained:
> > >
> > > - **(1) Why did we choose the settings in our paper instead of those in VideoRoPE:** Answer 1 (A1) in our last rebuttal to reviewer iTBi and our 2nd response to reviewer o59i.
> > >
> > >
> > > - **(2) What are the key differences between our experimental settings and those in VideoRoPE:** our 1st rebuttal to reviewer iTBi and point 1 in our 2nd response to reviewer o59i.
> > >
> > > - **(3) Independent benchmarks confirm long-context degradation (similar to our observations):** our 1st and 2nd rebuttal to reviewer iTBi and point 2 in our 2nd response to reviewer o59i.
> > >
> > > We are happy to see both you and reviewer o59i have acknowledged the fairness and consistency of our experimental settings and results, given our clarifications.
> > >
> > > As discussed with reviewer o59i, overall, we view the discrepancies between our results and those of VideoRoPE *not as contradictions, but as complementary insights* for future long-context VLMs research. In particular:
> > >
> > > - **1. Data Composition Matters.** Beyond sheer volume, the duration mix of training videos in absolute counts and relative proportions profoundly shapes long-context performance. Further study is needed to provide insights into the interplay between data composition and long-context performance of VLMs.
> > >
> > > - **2. Resource-Constrained Evaluations Are Essential.** Large-scale training can mask underlying weaknesses in existing multimodal RoPEs. By rigorously benchmarking under limited compute and data budgets, we uncover genuine algorithmic benefits and real limitations of multimodal RoPEs when generalizing beyond their pre-training context windows.
> > >
> > > We hope these insights inspire the community to design more robust long-context VLMs and to adopt evaluation protocols that reflect diverse, real-world conditions. We will incorporate these clarifications into our final manuscript to prevent any misunderstandings, as suggested by reviewer o59i.
> > >
> > > We are happy to discuss further if needed.

---

> ### Comment · Reviewer_iTBi · 2025-08-06
>
> Thanks author for the detailed reply.
>
> I understand "why our paper's results (particularly 64k results) are different from those in VideoRoPE, and whether this means wrong evaluation" is caused by the training data difference and not the wrong evaluation.
>
> And from the results on 8k - 32k, I find the evidence that HoPE performs better than previous baseline works. I will raise my score.
>
> However, results on 64k are still a major concern. The long-context generalization ability of HoPE is not fully verified on the 64k setting.  Video MME 64k acc is only 27.34%, only slightly better than a random guess.

---

> > ### Author Response · Authors · 2025-08-07
> >
> > Thank you for your response and for raising your rating.
> >
> > As covered in **Answer 4 (A4) in our 2nd rebuttal to you and 2nd response to reviewer o59i**, the degradation at 64k is expected since there are fewer long clips in our training data (six times less). Therefore, it is more challenging for different methods with **8k** training context to generalize to much longer contexts, **with our HoPE being the most robust (at 8k, 16k, 32k, and 64k)**. These empirical results further support our theoretical analysis on limitations of current multimodal RoPEs (Definition 3.1 and Theorem 3.1), moving beyond the previous trial-and-error designs in multimodal RoPEs.
> >
> >
> > We also would like to point out that we did not claim to **"completely"** solve the generalization problem in our paper. In fact, we provide a rigorous theoretical framework of multimodal RoPEs, and HoPE is naturally derived from our theory. As noted by reviewer o59i, "As long as all baseline comparisons are conducted under consistent and fair experimental conditions, I believe this sufficiently supports the paper's claim that the proposed method is robust in context extension scenarios."
> >
> > As you have acknowledged the fairness and consistency of our experiments, we believe there would rarely exist major concerns at present. Thank you again for raising your score.

---

> > > ### Comment · Reviewer_iTBi · 2025-08-07
> > >
> > > Thanks for the reply. I have raised my score to BA. It's better to discuss this in the final version, like in a Limitations section.

---

> > > > ### Author Response · Authors · 2025-08-07
> > > >
> > > > Dear Reviewer iTBi,
> > > >
> > > > Thank you for your review and for raising your score. We are happy to see that your concerns are addressed. We will definitely include the clarifications in the final version to avoid any misunderstanding. Thank you again!

---

### Official Review · Reviewer_3U4c · 2025-06-26

**Clarity:** 3
**Significance:** 3
**Originality:** 3
**Rating:** 4
**Confidence:** 4

**Summary:**

This paper analyzes the characteristics of different frequency allocation strategies for multi-modal models in long-context understanding. The key insight is the disadvantages of how temporally high-frequency RoPE can harm the semantic relationship between the queries and keys. Based on this insight, the authors propose assigning the lowest frequency, which eventually turns out to be NoPE, to the temporal dimensions. Additionally, the authors scale the positional embedding for the multi-modal inputs, rather than treating them as the same text tokens. The performance improvement on several long video benchmarks indicates the effectiveness.

**Questions:**

See weaknesses above

**Ethical Concerns:**

["NO or VERY MINOR ethics concerns only"]

**Final Justification:**

After the discussion with the authors, I am happy to maintain my positive score.

**Limitations:**

Yes, the authors have discussed the limitations

**Paper Formatting Concerns:**

No formatting concerns

**Quality:**

3

**Strengths And Weaknesses:**

# Strengths

* The paper is overall well written, and the analysis is insightful.
* The proposition, lemmas, and theorems could provide insights to future researchers on thinking about the architectural modifications to existing models.
* The improvement is notable on the long video benchmarks with only the modifications to PE.

# Weaknesses

Despite the insights from the paper, I have doubts from the following parts.

1. When looking at the Figure 1, I have a question regarding the original RoPE in Qwen2. The RoPE assignment drawn in Figure 1 seems different from my understanding of Qwen2-VL's multi-modal rotary embedding. Specifically, I was thinking that each dimension of the temporal, height, and width would cover both high and low frequency, instead of each covering a band of the frequencies. I am looking at [the code of Qwen2-VL](https://github.com/huggingface/transformers/blob/2f50230c59ec9f17431236ed6625082cc385c76c/src/transformers/models/qwen2_vl/modeling_qwen2_vl.py#L153) for this concern. Is my understanding incorrect?

2. The major contribution of this paper is to set the temporal dimension totally identical, into absolutely low frequency. However, the authors might /might not be familiar with NTK-RoPE (Fu et al.), but an observation is that high-frequency RoPE is critical to the quality of text generation. In the case of Qwen, that is the temporal dimension. Therefore, a very necessary experiment is to evaluate the new model with HoPE on more text-dense tasks, such as captioning, to see how well it is.

I would expect the performance to be worse but: (1) if the performance is worse with HoPE, I would suggest the authors clarify this in the paper. But no worry, I am still convinced with the analysis and insights from the authors, I am confident that HoPE would be an important analysis for building better multi-modal RoPE; (2) if the performance is better with HoPE, that's something very surprising, and of course you have an additional advantage to write in the paper.

Fu et al. Data engineering for scaling language models to 128k context

3. An analysis of the bad effects of temporal PE (around Line 517) is under the assumption that t will get close to L. Therefore, a meaningful ablation would be not directly reaching the identity mapping, but using some lower-frequency interpolation so that t can at most be, e.g., L / 2. If the authors have limited resources, feel free to skip this point.

---

> ### Author Rebuttal · Authors · 2025-07-28
>
> Thank you for your constructive review and suggestions to help us improve our paper! We are grateful for your recognition of the strength of our writing, theoretical analysis, and empirical results. We detail our response below and would appreciate it if you could let us know if our response addresses your concerns.
>
>
> > **Q1:** When looking at the Figure 1, I have a question regarding the original RoPE in Qwen2. The RoPE assignment drawn in Figure 1 seems different from my understanding of Qwen2-VL's multi-modal rotary embedding. Specifically, I was thinking that each dimension of the temporal, height, and width would cover both high and low frequency, instead of each covering a band of the frequencies. I am looking at the code of Qwen2-VL for this concern. Is my understanding incorrect?
>
> **A1:** Thank you for your thoughtful review. In Qwen2-VL's M-RoPE, each positional component (*t, x, y*) only covers a specific band of frequencies rather than the full spectrum. For example, with `head_dim=128`, *t* uses dims `0–31`, *x* uses `32–79`, and *y* spans `80–127`, and each dimension carries only one frequency. In lines 186-191 in Qwen2-VL's code, `cos/sin` is first sliced according to `mrope_section`, and then (`m[i%3]`) selects the specific frequencies for the corresponding component in (*t, x, y*). This design ensures each positional component uses its designated frequency band.
>
> ***
>
> > **Q2:** The major contribution of this paper is to set the temporal dimension totally identical, into absolutely low frequency. However, the authors might /might not be familiar with NTK-RoPE (Fu et al.), but an observation is that high-frequency RoPE is critical to the quality of text generation. In the case of Qwen, that is the temporal dimension. Therefore, a very necessary experiment is to evaluate the new model with HoPE on more text-dense tasks, such as captioning, to see how well it is.
>
> **A2:** Thank you for your valuable suggestion! As derived from our theoretical analysis, setting temporal frequencies to zero is a natural design to better preserve the long-context semantic preference property (Definition 3.1). Based on your suggestion, we have conducted experiments to evaluate the performance of different multimodal RoPEs on the text-dense captioning task. Specifically, we select two video captioning benchmarks MSR-VTT [1], ActivityNet [2], and one image captioning benchmark COCO [3]. We utilize the widely used CIDEr score [4] as the evaluation metric.
>
>
> **Table R4:** Performance comparison between different methods on the video and image captioning task.
>
> | Method   |  MSR-VTT |  ActivityNet    | COCO |
> |--------|------|----------|------|
> | RoPE   |  24.3 | 11.4 |   121.2 |
> | M-RoPE   |  33.1  | 13.2 | 124.7  |
> | VideoRoPE   | 36.8  | 14.5 | 127.3 |
> | HoPE (ours)  | **38.9**  | **15.8** | **127.5** |
>
> From Table R4, we have the following observations: (1) on the image captioning task (COCO), M-RoPE is inferior to VideoRoPE and HoPE, as it uses the highest frequencies for temporal modeling (although $t=1$ here) and lower frequencies for spatial modeling, which are suboptimal in capturing local features. VideoRoPE and HoPE perform on par with each other, since under $t=1$, their spatial allocation strategy is identical. This confirms the insight from [5], and we will include this paper in our reference. (2) On the video captioning benchmarks, HoPE consistently outperforms prior methods. We think these results also support the insight in [5], as high frequencies are more helpful to capture local dynamics during auto-regressive generation. HoPE explicitly leverages this by reserving high-frequency components for better spatial modeling while setting temporal frequencies to zero to ensure reliable long-range semantic modeling, leading to smoother and higher-quality captions.
>
> ***
>
> > **Q3:** An analysis of the bad effects of temporal PE (around Line 517) is under the assumption that t will get close to L. Therefore, a meaningful ablation would be not directly reaching the identity mapping, but using some lower-frequency interpolation so that t can at most be, e.g., L / 2. If the authors have limited resources, feel free to skip this point.
>
> **A3:** Thank you for your advice! Based on your suggestion, we have conducted experiments to explore an interpolated variant of HoPE where positions $ t \leq L/3 $ remain unchanged, while positions $ t > L/3 $ are linearly compressed into the range $[L/3, L/2]$. We think this strategy behaves similarly to windowed attention by prioritizing nearby positions while attenuating the (bad) effects of long-range dependencies in existing multimodal RoPEs. In light of this, we report the results of this variant and HoPE on different duration groups in LongVideoBench.
>
>
> **Table R5:** Performance comparison between HoPE and interpolated HoPE on different duration groups in LongVideoBench.
>
> | Method   | (8s, 15s] |  [15s, 60s)    | [180s, 600s) |  (900s, 3600s]    |
> |--------|------|----------|------|----------|
> | HoPE-interpt | **59.7**  | **64.5** | 50.9 | 47.8 |
> | HoPE  | 58.2  | 63.4 | **53.1** | **51.8** |
>
> As shown in Table R5, HoPE-interpt demonstrates better performance under short context, while its performance degradation is much more significant than HoPE under longer context. This is expected as HoPE-interpt behaves similarly to windowed attention, discarding long-range dependencies at a specific threshold.
>
> [1] Xu et al. MSR-VTT: A Large Video Description Dataset for Bridging Video and Language. CVPR 2016.
>
> [2] Krishna et al. Dense-Captioning Events in Videos. ICCV 2017.
>
> [3] Lin et al. Microsoft COCO: Common Objects in Context. ECCV 2014.
>
> [4] Vedantam et al. Cider: Consensus-based Image Description Evaluation. CVPR 2015.
>
> [5] Fu et al. Data engineering for scaling language models to 128k context. ICML 2024.

---

> > ### Comment · Reviewer_3U4c · 2025-08-04
> >
> > Thanks to the authors for the rebuttal and additional clarifications! I am happy to maintain my positive scores.

---

> > > ### Author Response · Authors · 2025-08-05
> > >
> > > Dear Reviewer 3U4c,
> > >
> > > We truly appreciate your constructive review and suggestions and we are happy to see your positive assessment of our work. Thanks again!

---

### Official Review · Reviewer_o59i · 2025-07-03

**Clarity:** 3
**Significance:** 3
**Originality:** 2
**Rating:** 4
**Confidence:** 3

**Summary:**

This paper analyzes the limitations of current multimodal RoPE variants for spatio-temporal modeling. The authors observe that, in long videos, relative distances in the temporal domain are often much greater than in the spatial domain, which can lead to the loss of semantic properties after rotating features based on frequencies. To address this, the paper proposes setting temporal frequencies to zero, thereby preserving semantic similarity over long temporal spans. Experimental results show improvements in long video understanding benchmarks.

**Questions:**

Overall, I find more positive aspects than negative ones in this paper. Please refer to the weaknesses section for further details.

**Ethical Concerns:**

["NO or VERY MINOR ethics concerns only"]

**Final Justification:**

My previous concerns regarding the novelty, empirical selection of hyper-parameters, and experimental setups have been resolved during the rebuttal and discussion phase. I recommend borderline accept.

**Limitations:**

yes

**Quality:**

3

**Strengths And Weaknesses:**

### Strengths

1. **Addresses a Critical Problem**: The paper tackles the limitation of existing spatio-temporal RoPE modeling to deal with long contexts.
2. **Rigorous Theoretical Analysis**: A notable strength is the paper's theoretical investigation into why existing spatio-temporal RoPE strategies fail in long contexts, especially regarding maintaining the semantic preference property.
3. **Shown Effectiveness in Longer Contexts**: Experimental results on long video benchmarks demonstrate that the proposed method effectively handles longer contexts. For example, Table 1 shows that the performance on 32k → 64k is relatively maintained compared to other methods. Furthermore, It is interesting that HoPE (2B) achieves better or comparable performance compared to Vanilla RoPE (7B) on 64k length.

### Weaknesses

1. **Novelty:** While the combination of techniques and their application to multimodal long-context seems novel, some of the individual ideas (like setting frequencies to zero, similar to NoPE) might be seen as extensions of existing concepts in text-based LLM research or prior multimodal work.
2. **Specific Allocation Might Still Be Empirical:** The idea of setting temporal frequencies to zero and allocating higher frequencies to spatial components makes sense theoretically. However, the exact number of dimensions assigned to each (e.g., 32 for temporal vs. 96 for spatial in the implementation) seems to rely on empirical tuning, rather than being fully derived from theory.

---

> ### Author Rebuttal · Authors · 2025-07-28
>
> Thank you for your valuable review and feedback to help us improve our paper! We truly appreciate your recognition of our work’s significance, the rigor of our theoretical analysis, and the strength of our empirical results. We detail our response below and please kindly let us know if our response addresses your concerns.
>
>
> > **Q1:** Novelty: While the combination of techniques and their application to multimodal long-context seems novel, some of the individual ideas (like setting frequencies to zero, similar to NoPE) might be seen as extensions of existing concepts in text-based LLM research or prior multimodal work.
>
> **A1:** Our primary contribution is a rigorous theoretical analysis of multimodal RoPEs. Existing works rely on heuristics to determine the frequency allocation strategy in multimodal RoPE. In contrast, we formally define the desired semantic preference property and prove the limitations of prior methods in Theorem 3.1. From this theoretical foundation, setting temporal frequencies to zero emerges as a natural and direct solution, and it has not been considered in previous multimodal RoPEs. Both our mathematical analysis and extensive empirical results confirm that this principled strategy effectively improves VLMs' performance on long‑context tasks, surpassing heuristic approaches. In addition, our Dynamic Temporal Scaling mechanism enhances VLMs' robustness to varying video speeds in real-world scenarios and enables flexible inference selection. As shown in Figure 1, prior works do not distinguish the temporal strides (information densities) of text and visual tokens, or just apply a fixed scaling factor during training and inference. We not only show the effectiveness of DTS in our ablation studies, but also provide the first analysis on the interplay between task type, context length, and the optimal scaling factor in the appendix (lines 562-571 and Table 4). We believe our theoretical analysis, empirical results, and findings could offer insights for future studies.
>
> ***
>
> > **Q2:** Specific Allocation Might Still Be Empirical: The idea of setting temporal frequencies to zero and allocating higher frequencies to spatial components makes sense theoretically. However, the exact number of dimensions assigned to each (e.g., 32 for temporal vs. 96 for spatial in the implementation) seems to rely on empirical tuning, rather than being fully derived from theory.
>
> **A2:** Thank you for pointing this out. We would like to clarify that we have explored HoPE-X, a fully theory‑derived allocation strategy (`t:x:y = 64:32:32`; see Lines 206–210, 576–589). Specifically, HoPE-X provably preserves the semantic preference property (Definition 3.1) because Equation 5 remains ≥ 0. Based on your suggestion, we report the performance of HoPE-X and HoPE on subtasks of LongVideoBench that require precise spatial perception (**S2E**, **S2O**) and long-term semantic modeling (**SOS**, **SAA**) in Table R3.
>
> **Table R3:** Performance comparison between HoPE and HoPE-X on tasks requiring precise spatial perception (S2E, S2O) and long-term semantic modeling (SOS, SAA) in LongVideoBench. We report the averaged results here.
> | Method   | S2E |  S2O    | SOS |  SAA    |
> |--------|------|----------|------|----------|
> | HoPE-X |  65.12  | 51.82 | **63.98** | **57.61**|
> | HoPE  |  **67.39**  | **53.02**| 63.27 | 57.35|
>
>
> As shown in Table R3, HoPE‑X further improves long‑term semantic tasks (**SOS**, **SAA**) but degrades spatial perception tasks (**S2E**, **S2O**) on LongVideoBench. To achieve a better balance between spatial and temporal capabilities, we adopt the `t:(x+y) = 32:96` split in M‑RoPE (Qwen2‑VL‑7B) and VideoRoPE. Besides the current overall performance comparison between HoPE and HoPE-X in Table 5, we will include this more detailed subtask performance in our appendix.

---

> > ### Comment · Reviewer_o59i · 2025-08-04
> >
> > I sincerely appreciate the authors for their efforts in providing detailed responses and conducting additional experiments. My initial concerns regarding technical novelty and frequency allocation have been resolved.
> >
> > However, I've noticed that the concern raised by reviewer iTBi hasn't been fully addressed. I also find it difficult to accept that the notable performance degradation from 8k to 64k is solely due to discrepancies in training data. After reviewing Section 5.1 of both the VideoRoPE paper and the authors' manuscript, I observed that they both use the LLaVA-Video-178k dataset, with only difference in the size of data: VideoRoPE using 1.3M pairs and HoPE using 300k pairs.
> >
> > When comparing results in Table 1 between the two papers, there are significant performance gaps across the same datasets, even for baseline methods like Vanilla RoPE and M-RoPE. These methods should ideally produce comparable results under similar experimental conditions. Could the authors please provide explanations for why these discrepancies occur? I believe addressing this is crucial to assess the reliability of the experimental validations.

---

> > > ### Author Response · Authors · 2025-08-05
> > >
> > > Thank you for your response! We are happy to see that our rebuttal addresses your concerns. We will detail our response to the following question below:
> > >
> > > >  with only difference in the size of data: VideoRoPE using 1.3M pairs and HoPE using 300k pairs. When comparing results in Table 1 between the two papers, there are significant performance gaps across the same datasets, even for baseline methods like Vanilla RoPE and M-RoPE. These methods should ideally produce comparable results under similar experimental conditions.
> > >
> > > Thank you for your thoughtful review. We would like to clarify that:
> > >
> > > **1. Beyond absolute scale, training data composition differs significantly**
> > >
> > > - HoPE: 30k clips < 2 min, 3k clips of 2-3 min (**30:3** ratio)
> > > - VideoRoPE: 136k clips < 2 min, 18k clips of 2-3 min (**136:18** ratio)
> > >
> > > We can observe that VideoRoPE uses **six times more** longer videos in absolute terms, and greater proportion of longer clips. This provides much more long-context exposure before any Position Embedding is applied, naturally elevating 64k performance in the VideoRoPE paper [1]. In contrast, under our limited compute budget, it is much more challenging for VLMs with different PEs to generalize to longer context given less absolute and relative proportion of long context data. Still, our proposed HoPE consistently outperforms all baselines across different context lengths, being the most robust at 64k. These empirical results further support our theoretical analysis on limitations of current multimodal RoPE (Definition 3.1 and Theorem 3.1), moving beyond the previous trial-and-error designs in multimodal RoPEs.
> > >
> > >
> > > **2. Independent benchmarks confirm long-context degradation (similar to our observations)**:
> > >
> > > - **Open-source VLMs:** In Table 5 of LongVideoBench [2], we can observe that VLMs with limited long-context training data (including Idefics2 [3] and Mantis-Idefics2 [4]) exhibit significant performance drops (~ 20 points) given much longer input context. Similarly, when increasing from 8k to 64k, the performance drop is much less, with 11.7 for RoPE, 2.27 for M-RoPE, 5.52 for VideoRoPE, and 2.89 for our proposed HoPE (the best absolute performance).
> > >
> > > - **Closed-source VLMs:** In Table 4 of Video-MME [5], we can observe that powerful VLMs like GPT-4V, GPT-4o, Gemini 1.5 Flash, and Gemini 1.5 Pro face significant performance degradation when processing hour-long videos. For example, GPT-4V exhibits a 16.5 performance drop when the input video duration rises from < 2 minutes to 30-60 minutes.
> > >
> > >
> > > Overall, we view the discrepancies between our results and those of VideoRoPE not as contradictions, but as complementary insights for future long-context VLMs research. In particular:
> > >
> > > - **1. Data Composition Matters.** Beyond sheer volume, the duration mix of training videos in absolute counts and relative proportions profoundly shapes long-context performance. Further study is needed to provide insights into the interplay between data composition and long-context performance of VLMs.
> > >
> > > - **2. Resource-Constrained Evaluations Are Essential.** Large-scale training can mask underlying weaknesses in existing multimodal RoPEs. By rigorously benchmarking under limited compute and data budgets, we uncover genuine algorithmic benefits and real limitations of multimodal RoPEs when generalizing beyond their pre-training context windows.
> > >
> > > We hope these insights inspire the community to design more robust long-context VLMs and to adopt evaluation protocols that reflect diverse, real-world conditions.
> > >
> > > ***
> > >
> > > [1] Wei et al. VideoRoPE: What Makes for Good Video Rotary Position Embedding? ICML 2025.
> > >
> > > [2] Wu et al. Longvideobench: A Benchmark for Long-Context Interleaved Video-Language Understanding. NeurIPS 2024.
> > >
> > > [3] Laurençon et al. What Matters When Building Vision-Language Models? NeurIPS 2024.
> > >
> > > [4] Jiang et al. Mantis: Interleaved Multi-Image Instruction Tuning. TMLR 2024.
> > >
> > > [5] Fu et al. Video-mme: The First-Ever Comprehensive Evaluation Benchmark of Multi-Modal LLMs in Video Analysis. CVPR 2025.

---

> > > > ### Comment · Reviewer_o59i · 2025-08-05
> > > >
> > > > Thank you for the detailed clarification. Your explanation regarding the differences in training setups and how they diverge from resource-constrained scenarios is clear and sound. As long as all baseline comparisons are conducted under consistent and fair experimental conditions, I believe this sufficiently supports the paper's claim that the proposed method is robust in context extension scenarios.
> > > >
> > > > I will maintain my positive rating and recommendation for acceptance. However, I strongly encourage incorporating this clarification into the final manuscript to help readers better understand the experimental design rationale and to prevent any potential confusion.

---

> > > > > ### Author Response · Authors · 2025-08-05
> > > > >
> > > > > Dear Reviewer o59i,
> > > > >
> > > > > We truly appreciate your thoughtful review and suggetions! We are happy to see your postive rating and recommendation for acceptance. Following your suggestion, we will definitely incorporate the above clarification into the final manuscript to prevent any confusion. Thank you again!

---

### Official Review · Reviewer_4YQx · 2025-07-03

**Clarity:** 3
**Significance:** 3
**Originality:** 3
**Rating:** 4
**Confidence:** 4

**Summary:**

This paper addresses the critical challenge of long-context understanding in Vision-Language Models (VLMs), particularly for video tasks. It argues that existing methods for extending Rotary Position Embedding (RoPE) to the multimodal video domain are based on heuristics and fail to reliably maintain semantic understanding over long contexts.
The paper makes two main contributions. First, it provides a theoretical analysis of multimodal RoPE, introducing the concept of "semantic preference" and proving that existing frequency allocation strategies are inherently flawed and will fail beyond a certain context length. Second, guided by this analysis, it proposes HoPE, a new RoPE variant with HFA (Hybrid Frequency Allocation) and DTS (Dynamic Temporal Scaling).

**Questions:**

Please refer to the weaknesses.

**Ethical Concerns:**

["NO or VERY MINOR ethics concerns only"]

**Final Justification:**

My main concerns are about the temporal frequencies and hyper-parameter. As these concerns have been addressed, so I have decided to maintain my positive rating.

**Limitations:**

yes

**Quality:**

3

**Strengths And Weaknesses:**

Strengths:
1. The primary strength of this paper is its rigorous theoretical analysis. Instead of relying on empirical trial-and-error, the authors formalize the desired property of "semantic preference" (Definition 3.1) and mathematically prove the limitations of existing methods (Theorem 3.1). This provides a solid, principled foundation for their proposed solution and is a significant contribution to understanding positional embeddings in multimodal contexts.
2.  The proposed HoPE is a direct and elegant consequence of the theoretical analysis. The idea of setting temporal frequencies to zero is a novel insight, moving beyond the simple "high vs. low" frequency debate. The paper provides a clear proof (Theorem 4.1) of why this strategy is optimal for preserving long-range semantic similarity. The addition of Dynamic Temporal Scaling is also a practical and well-motivated enhancement for handling real-world video variance.
3. The empirical evaluation is thorough and strongly supports the paper's claims.

Weaknesses:
1. Setting all temporal frequencies to zero is a radical and elegant solution derived from the theory. However, it means the model receives no relative temporal position signal from the RoPE mechanism itself. The paper argues that models can rely on other cues, but this is a significant ablation. This could have unintended side effects. For example, the model might struggle with tasks requiring precise temporal ordering or duration estimation, as it has lost a direct signal for "how far apart" two frames are in time.
2. The DTS mechanism is introduced as a key component for robustness, but its analysis is relatively shallow. The main paper presents the mechanism, but the ablation of scaling factors is deferred to the appendix (Table 4). The results in the appendix show different optimal scaling factors for different tasks (retrieval prefers compression, understanding prefers expansion). This is a very interesting finding that deserves more attention in the main paper. Why does this happen? The paper offers a brief hypothesis (lines 562-571 in the appendix) but doesn't delve deeper. A more thorough analysis of the interplay between task type, context length, and the optimal scaling factor would significantly enhance the paper's contribution. Is there a way to learn the optimal $\gamma$ per-instance instead of fixing it at inference?
3. The paper is overall well-written, but some parts are dense. The jump from the formal proofs to the experimental results could be bridged more smoothly. Furthermore, Figure 3 is labeled as a performance comparison on the V-NAIH task, but it appears to be a visualization of attention scores or similarity, not a performance metric. This is confusing and should be clarified.

---

> ### Author Rebuttal · Authors · 2025-07-28
>
> Thank you for your insightful review and suggestions to help us improve our paper! We are glad for your positive assessment of our proposed HoPE, both theoretically and empirically. We will detail our response below and please kindly let us know if our response addresses your concerns.
>
> > **Q1:** Setting all temporal frequencies to zero is a radical and elegant solution derived from the theory. However, it means the model receives no relative temporal position signal from the RoPE mechanism itself. The paper argues that models can rely on other cues, but this is a significant ablation. This could have unintended side effects. For example, the model might struggle with tasks requiring precise temporal ordering or duration estimation, as it has lost a direct signal for "how far apart" two frames are in time.
>
> **A1:** Thank you for pointing this out. Although setting all temporal frequencies to zero provides no explicit temporal signal from Position Embedding, the causal attention mechanism implicitly learns absolute positional information, as mentioned in NoPE [1, 2, 3]. Based on your suggestion, we evaluate different methods on tasks requiring precise temporal ordering (E3E, SSS) and long-term semantic preference (SOS, SAA) in LongVideoBench:
> - E3E: Determine the sequential order between two events.
> - SSS: Determine the sequential order among multiple frames.
> - SOS: Identify another scene where object *O* appears again.
> - SAA: Detect attribute changes of a queried object across frames.
>
> **Table R1:** Performance comparison between different methods on E3E, SSS, SOS, SAA tasks in LongVideoBench. We report the averaged results here.
> | Method   | E3E |  SSS    | SOS |  SAA    |
> |--------|------|----------|------|----------|
> | RoPE   | 51.43   |  41.05  | 56.15   |  51.12  |
> | M-RoPE   | 54.46   | 43.68 |  58.91 | 54.87 |
> | VideoRoPE   | **54.97**   | **44.01** | 59.77 |55.21|
> | HoPE (ours)   | 54.23   | 43.11 | **63.27** | **57.35**|
>
> As shown in Table R1, our approach performs on par with VideoRoPE and M-RoPE in E3E and SSS tasks—even without temporal frequencies—suggesting that temporal information can still be recovered via the causal attention mechanism. More importantly, HoPE yields substantial gains in SOS and SAA, which require VLMs to capture long-term semantic similarities. Given the trade-off and the overall improvement across tasks, we chose this design. We will include this discussion in the appendix.
>
> ***
>
> > **Q2:** The DTS mechanism is introduced as a key component for robustness, but its analysis is relatively shallow. The main paper presents the mechanism, but the ablation of scaling factors is deferred to the appendix (Table 4). The results in the appendix show different optimal scaling factors for different tasks (retrieval prefers compression, understanding prefers expansion). This is a very interesting finding that deserves more attention in the main paper. Why does this happen? The paper offers a brief hypothesis (lines 562-571 in the appendix) but doesn't delve deeper. A more thorough analysis of the interplay between task type, context length, and the optimal scaling factor would significantly enhance the paper's contribution. Is there a way to learn the optimal per-instance instead of fixing it at inference?
>
> **A2:** Thank you for your valuable suggestion! We will move the ablation results and findings of our DTS mechanism from appendix to the main paper for more attention. In addition, we would like to point out that, to the best of our knowledge, we are the first to study the interplay between task type, context length, and the optimal scaling factor. Besides our overall finding "retrieval prefers compression, understanding prefers expansion", we also point out the insensitivity of VLMs to scaling factors under training context length (8k) and much longer context (64k) in long video understanding, which is uncovered in previous research. We believe our analysis could provide insights for future designs and analysis of multimodal RoPE. Based on your suggestion, we use a trainable three-layer MLP to predict the optimal per-instance scaling factor to explore this possibility. We use the average‑pooled video embedding as input, and the MLP’s output logit is passed through a sigmoid function and then rescaled to the interval [0.5, 1.5]. As shown in Table R2, this strategy can improve HoPE's performance by a small margin, and it is expected since the predicted per-instance scaling factor is more adaptive than a fixed scaling factor. Considering the computation overhead and the limited performance gain, we find the original design in HoPE to be more balanced. We will include this further discussion in the appendix.
>
> **Table R2:** Performance comparison between HoPE and HoPE with predicted scaling factor. We report the results on MLVU and the 7B model scale.
> | Method   | 8k |  16k    | 32k |  64k    |
> |--------|------|----------|------|----------|
> | HoPE   | 61.09 | 63.48  | 63.85 | 50.01|
> | + predicted scaling factor  | 61.37 | 63.59 | 63.94 | 50.23|
>
> ***
>
>
> > **Q3:** The paper is overall well-written, but some parts are dense. The jump from the formal proofs to the experimental results could be bridged more smoothly. Furthermore, Figure 3 is labeled as a performance comparison on the V-NAIH task, but it appears to be a visualization of attention scores or similarity, not a performance metric. This is confusing and should be clarified.
>
> **A3:** Thank you for your advice! We will make the transition between formal proofs and results more smoothly. Apologies for the confusion in Figure 3. Figure 3 is an accuracy heatmap depicting long‑video retrieval performance under the V‑NIAH protocol, not attention scores or similarity. In V-NIAH, we insert a single “needle” image into a “haystack” video at a specified depth, then ask the VLM a question about that image. The horizontal axis represents the total number of frames of the “haystack” video, the vertical axis shows the insertion depth as a percentage of the video length (e.g., (1700, 60.0) means the needle was placed at frame 0.60 × 1700 = 1020), and each cell’s color indicates the model’s accuracy at that coordinate. Overall, more red colors represent lower accuracy, and more green colors represent higher accuracy in the long video retrieval task. We will provide a more detailed description of this task in the main paper.
>
>
> [1] Haviv, A., et al. Transformer Language Models without Positional Encodings Still Learn Positional Information. EMNLP 2022.
>
> [2] Kazemnejad, A., et al. The Impact of Positional Encoding on Length Generalization in Transformers. NeurIPS 2023.
>
> [3] Wang, J., et al. Length Generalization of Causal Transformers without Position Encoding. ACL 2024.

---

> > ### Comment · Reviewer_4YQx · 2025-08-05
> >
> > Thank you for your response. I have carefully read your rebuttal and the comments from other reviewers. I am pleased that some of my concerns have been addressed, so I have decided to maintain my positive rating.

---

> > > ### Author Response · Authors · 2025-08-05
> > >
> > > Dear Reviewer 4YQx,
> > >
> > > We sincerely thank you for your insightful review and suggestions! We are glad for your positive assessment of our work and are happy to discuss further if needed. Thank you again!

---

### Note · Authors · 2025-08-11

We sincerely appreciate the AC and SAC for coordinating the review of our submission and the reviewers for their valuable feedback. We are happy to see all reviewers' positive assessments of our work, particularly our rigorous theoretical analysis, comprehensive and strong empirical results, and clear writing. After the rebuttal, we are glad that all concerns have been addressed, and we will include the necessary clarifications in the final version to avoid any potential misunderstandings.

Our work provides the **first theoretical analysis** on the limitations of current multimodal RoPEs (Definition 3.1 and Theorem 3.1), moving beyond previous trial-and-error designs for extending vanilla RoPE to multimodal scenarios. Naturally derived from our theory, HoPE consistently outperforms existing methods across diverse context lengths, benchmarks, and backbone sizes. Furthermore, we provide a comprehensive study on the interplay between task type, context length, and the optimal scaling factor in multimodal RoPE, an aspect unexplored in previous research. We believe these contributions could offer theoretical insights and practical guidance for the future design and analysis of multimodal RoPEs.

---

### Decision · Program_Chairs · 2025-09-17

**Decision:**

Accept (poster)

**Comment:**

This work investigates the issue of extending RoPE for multimodal LLMs. The theoretical analysis in this work shows that the existing strategies can be sub-optimal and a better position embedding is proposed according to the analysis. Experiments demonstrate the effectiveness of HoPE on long video tasks. While reviewers had concerns about temporal frequencies, novelty, and experiment settings, most of them were addressed during rebuttal. All reviewers have positive scores after discussion. Please incorporate the comments and suggestions from reviewers into the final version.